# FUS regulates RAN translation through modulating the G-quadruplex structure of GGGGCC repeat RNA in *C9orf72*-linked ALS/FTD

Yuzo Fujino[1,2], Morio Ueyama[1,3,4], Taro Ishiguro[4,5], Daisaku Ozawa[1,3], Hayato Ito[6], Toshihiko Sugiki[7], Asako Murata[8], Akira Ishiguro[9], Tania Gendron[10], Kohji Mori[11], Eiichi Tokuda[12], Tomoya Taminato[1,3], Takuya Konno[13], Akihide Koyama[13], Yuya Kawabe[11], Toshihide Takeuchi[3,14], Yoshiaki Furukawa[12], Toshimichi Fujiwara[7], Manabu Ikeda[11], Toshiki Mizuno[2], Hideki Mochizuki[15], Hidehiro Mizusawa[5], Keiji Wada[4], Kinya Ishikawa[5], Osamu Onodera[13], Kazuhiko Nakatani[8], Leonard Petrucelli[10], Hideki Taguchi[6,16], Yoshitaka Nagai[1,3,4,15]*

[1]Department of Neurology, Kindai University Faculty of Medicine, Osaka-Sayama, Japan; [2]Department of Neurology, Kyoto Prefectural University of Medicine, Kyoto, Japan; [3]Department of Neurotherapeutics, Osaka University Graduate School of Medicine, Suita, Japan; [4]Department of Degenerative Neurological Diseases, National Institute of Neuroscience, National Center of Neurology and Psychiatry, Tokyo, Japan; [5]Department of Neurology and Neurological Science, Tokyo Medical and Dental University, Tokyo, Japan; [6]School of Life Science and Technology, Tokyo Institute of Technology, Yokohama, Japan; [7]Laboratory of Molecular Biophysics, Institute for Protein Research, Osaka University, Osaka, Japan; [8]Department of Regulatory Bioorganic Chemistry, The Institute of Scientific and28 Industrial Research, Osaka University, Osaka, Japan; [9]Research Center for Micro-nano Technology, Hosei University, Tokyo, Japan; [10]Department of Neuroscience, Mayo Clinic, Jacksonville, United States; [11]Department of Psychiatry, Osaka University Graduate School of Medicine, Osaka, Japan; [12]Department of Chemistry, Keio University, Kanagawa, Japan; [13]Department of Neurology, Clinical Neuroscience Branch, Brain Research Institute, Niigata University, Niigata, Japan; [14]Life Science Research Institute, Kindai University, Osaka, Japan; [15]Department of Neurology, Osaka University Graduate School of Medicine, Osaka, Japan; [16]Cell Biology Center, Institute of Innovative Research, Tokyo Institute of Technology, Kanagawa, Japan

*For correspondence:
yoshi.nagai@med.kindai.ac.jp

**Abstract** Abnormal expansions of GGGGCC repeat sequence in the noncoding region of the *C9orf72* gene is the most common cause of familial amyotrophic lateral sclerosis and frontotemporal dementia (C9-ALS/FTD). The expanded repeat sequence is translated into dipeptide repeat proteins (DPRs) by noncanonical repeat-associated non-AUG (RAN) translation. Since DPRs play central roles in the pathogenesis of C9-ALS/FTD, we here investigate the regulatory mechanisms of RAN translation, focusing on the effects of RNA-binding proteins (RBPs) targeting GGGGCC repeat RNAs. Using C9-ALS/FTD model flies, we demonstrated that the ALS/FTD-linked RBP FUS suppresses RAN translation and neurodegeneration in an RNA-binding activity-dependent manner. Moreover, we found that FUS directly binds to and modulates the G-quadruplex structure of GGGGCC repeat RNA as an RNA chaperone, resulting in the suppression of RAN translation in vitro. These results reveal a

previously unrecognized regulatory mechanism of RAN translation by G-quadruplex-targeting RBPs, providing therapeutic insights for C9-ALS/FTD and other repeat expansion diseases.

## eLife assessment

This **important** study demonstrates that the human FUS protein, which is implicated in ALS and related conditions, interacts with RNAs containing GGGGCC repeats and can regulate their translation by altering three-dimensional structures caused by these repeats. The study is carefully executed and the data provide **convincing** evidence for its major claims. This work will likely be of interest to researchers studying RNA binding proteins, and to those working on ALS and related diseases.

## Introduction

Amyotrophic lateral sclerosis (ALS) and frontotemporal dementia (FTD) are incurable neurodegenerative diseases with overlapping genetic and neuropathological features. Abnormal expansions of the GGGGCC ($G_4C_2$) repeat sequence in the noncoding region of the *C9orf72* gene have been found to be the most common genetic mutation responsible for ALS/FTD (*DeJesus-Hernandez et al., 2011*; *Gijselinck et al., 2012*; *Renton et al., 2011*). Three major pathomechanisms are thought to be involved in the pathogenesis of *C9orf72*-linked ALS/FTD (C9-ALS/FTD): first, expansion of the $G_4C_2$ repeats results in decreased expression of the *C9orf72* gene, leading to its haploinsufficiency (*Boivin et al., 2020*; *DeJesus-Hernandez et al., 2011*; *Gijselinck et al., 2012*; *Shi et al., 2018*; *Waite et al., 2014*; *Zhu et al., 2020*). Second, the transcribed $G_4C_2$ repeat-containing RNA accumulates as RNA foci in the affected tissues, sequestering various RNA-binding proteins (RBPs) and altering their function (*Conlon et al., 2016*; *Cooper-Knock et al., 2014*; *Donnelly et al., 2013*; *Haeusler et al., 2014*; *Lee et al., 2013*; *Mori et al., 2013a*). Third, this $G_4C_2$ repeat RNA is also translated into dipeptide repeat (DPR) proteins, despite the lack of an AUG initiation codon, by noncanonical repeat-associated non-AUG (RAN) translation (*Ash et al., 2013*; *Gendron et al., 2013*; *Mori et al., 2013b*; *Mori et al., 2013c*; *Zu et al., 2011*; *Zu et al., 2013*). Since RAN translation occurs in all reading frames and the expanded $G_4C_2$ repeat sequence is bidirectionally transcribed, five distinct DPRs, that is, poly(glycine-arginine) [poly(GR)], poly(glycine-alanine) [poly(GA)], poly(glycine-proline) [poly(GP)], poly(proline-arginine) [poly(PR)], and poly(proline-alanine) [poly(PA)], are produced and observed in patients' brains (*Ash et al., 2013*; *Gendron et al., 2013*; *Mori et al., 2013b*; *Mori et al., 2013c*; *Zu et al., 2013*) and cerebrospinal fluid (*Gendron et al., 2017*; *Krishnan et al., 2022*; *Lehmer et al., 2017*; *Su et al., 2014*).

DPRs have been shown to exert toxic effects in multiple C9-ALS/FTD models, such as cultured cells, flies, and mice (*Choi et al., 2019*; *Jovičić et al., 2015*; *May et al., 2014*; *Mizielinska et al., 2014*; *Rudich et al., 2017*; *Wen et al., 2014*; *Zhang et al., 2016*; *Zhang et al., 2018*). Importantly, the toxicity of DPRs was confirmed in DPR-only flies, which express DPRs translated from non-$G_4C_2$ repeat RNAs with alternative codons and show neurodegeneration, whereas RNA-only flies expressing $G_4C_2$ repeat RNAs with stop codon interruptions, which eliminate DPRs production, did not show any obvious degenerative phenotypes (*Mizielinska et al., 2014*). In addition, increased DPR production, but not RNA foci, was reported to correlate with $G_4C_2$ repeat-induced toxicity in a C9-ALS/FTD *Drosophila* model (*Tran et al., 2015*). Taken together, these studies have strongly suggested that DPRs play a central role in the pathogenesis of C9-ALS/FTD. Indeed, DPRs have been reported to disrupt various biological pathways, such as nucleocytoplasmic transport (*Hutten et al., 2020*; *Jovičić et al., 2015*; *Zhang et al., 2016*) and membraneless organelle dynamics (*Kwon et al., 2014*; *Lee et al., 2016*; *Lin et al., 2016*). Therefore, elucidating the regulatory mechanism of RAN translation is a significant challenge toward developing potential therapies for C9-ALS/FTD.

Since the discovery of RAN translation in 2011 (*Zu et al., 2011*), many studies to date have focused on its molecular mechanisms, that is, whether it has functional overlap with canonical AUG-dependent translation. Previous studies on C9-ALS/FTD using monocistronic reporters containing a $G_4C_2$ repeat sequence revealed cap-dependent translation initiation from the upstream near-cognate CUG initiation codon, requiring the cap-binding eukaryotic translation factor 4F complex (*Green et al., 2017*; *Tabet et al., 2018*). On the other hand, studies using bicistronic reporters with a $G_4C_2$ repeat sequence

in the second cistron also produced DPRs by RAN translation in all reading frames, suggesting cap-independent translation initiation within the $G_4C_2$ repeat sequence (*Cheng et al., 2018*; *Sonobe et al., 2018*). This is reminiscent of internal ribosomal entry site translation initiation, which is another type of noncanonical cap-independent translation in which specific factors are directly recruited to the highly structured mRNA for initiation (*Kwan and Thompson, 2019*). While such initiation mechanisms of RAN translation have been explored to date, specific roles of the repeat sequence on RAN translation remain enigmatic. Considering a repeat length dependency of RAN translation (*Mori et al., 2013c*; *Zu et al., 2011*; *Zu et al., 2013*), the repeat sequence itself would also be essential for the initiation or elongation steps of RAN translation. Based on our previous findings of the protective role of TDP-43 on UGGAA repeat-induced toxicity in spinocerebellar ataxia type 31 (SCA31) models (*Ishiguro et al., 2017*), we hypothesized that RBPs specifically binding to repeat sequences of template RNA play a role in RAN translation.

Using *Drosophila* models of C9-ALS/FTD, we here demonstrate the regulatory roles of the ALS/FTD-linked RBP FUS on RAN translation from $G_4C_2$ repeat RNA, which lead to the significant modulation of neurodegeneration. We found that FUS suppresses RNA foci formation and DPR production, resulting in the suppression of repeat-induced degeneration. This suppressive effect on degeneration was abolished by mutations in the RNA-recognition motif (RRM) of FUS. In contrast, knockdown of endogenous *caz*, a *Drosophila* homologue of *FUS*, enhanced DPR aggregation and RNA foci formation, resulting in the enhancement of repeat-induced degeneration. Moreover, FUS was found to directly bind to $G_4C_2$ repeat RNA and modify its G-quadruplex structure as an RNA chaperone, resulting in the suppression of RAN translation in vitro. In addition, other G-quadruplex-targeting RBPs also suppressed RAN translation and $G_4C_2$ repeat-induced toxicity in our C9-ALS/FTD flies. These results strongly indicate that FUS regulates RAN translation and suppresses DPR toxicity through modulating the G-quadruplex structure of $G_4C_2$ repeat RNA. Our findings shed light on the regulatory mechanisms of RAN translation by G-quadruplex-targeting RBPs and propose novel therapeutic strategies for repeat expansion diseases by regulating RAN translation.

## Results

### Screening for RBPs that suppress $G_4C_2$ repeat-induced toxicity in C9-ALS/FTD flies

We established *Drosophila* models of C9-ALS/FTD that express pathogenic length 42 or 89 $G_4C_2$ repeats [$(G_4C_2)_{42}$, or $(G_4C_2)_{89}$ flies, respectively] and confirmed that expanded $G_4C_2$ repeat sequences induce eye degeneration and motor dysfunction accompanied with the formation of RNA foci and the production of three types of DPRs (*Figure 1—figure supplement 1*), consistent with previous studies (*Freibaum et al., 2015*; *Goodman et al., 2019*; *Mizielinska et al., 2014*; *Xu et al., 2013*). We also established *Drosophila* expressing normal length 9 $G_4C_2$ repeats as a control [$(G_4C_2)_9$ flies] and found that they did not show eye degeneration, motor dysfunction, RNA foci formation, or DPR aggregation (*Figure 1—figure supplement 1*). We selected 18 RBPs that have been reported to bind to $G_4C_2$ repeat RNA (*Mori et al., 2013a*), as well as TDP-43, an ALS/FTD-linked RBP that does not bind to $G_4C_2$ repeat RNA (*Xu et al., 2013*; *Figure 1—source data 1*), and examined their roles in neurodegeneration in our C9-ALS/FTD fly models. We found that coexpression of *FUS*, *IGF2BP1*, or *hnRNPA2B1* strongly suppressed the eye degeneration in both flies expressing $(G_4C_2)_{42 \text{ or } 89}$, which show decreased eye size and loss of pigmentation (*Figure 1A–D* and *Figure 1—source data 2*). Coexpression of five RBPs, namely, *hnRNPR*, *SAFB2*, *SF3B3*, *hnRNPA1*, and *hnRNPL*, also partially suppressed the eye degeneration, whereas coexpression of the other six RBPs had no effect, and two RBPs enhanced the phenotypes (*Figure 1A–D* and *Figure 1—source data 2*). In addition, coexpression of *TDP-43* had no effect on the eye degeneration in $(G_4C_2)_{42}$ flies and resulted in lethality in $(G_4C_2)_{89}$ flies, likely due to the toxicity of *TDP-43* expression itself (*Figure 1A and D* and *Figure 1—source data 2*). The variation in the effects of these $G_4C_2$ repeat-binding RBPs on $G_4C_2$ repeat-induced toxicity may be due to their different binding affinities to $G_4C_2$ repeat RNA and the different toxicity of overexpressed RBPs themselves. We then analyzed the expression levels of $G_4C_2$ repeat RNA in flies coexpressing $(G_4C_2)_{89}$ and three RBPs that strongly suppressed eye degeneration. We found that coexpression of *IGF2BP1* or *hnRNPA2B1* significantly decreased $G_4C_2$ repeat RNA levels, whereas they were not altered upon coexpression of *FUS* (*Figure 1E*). Although the suppressive effects of

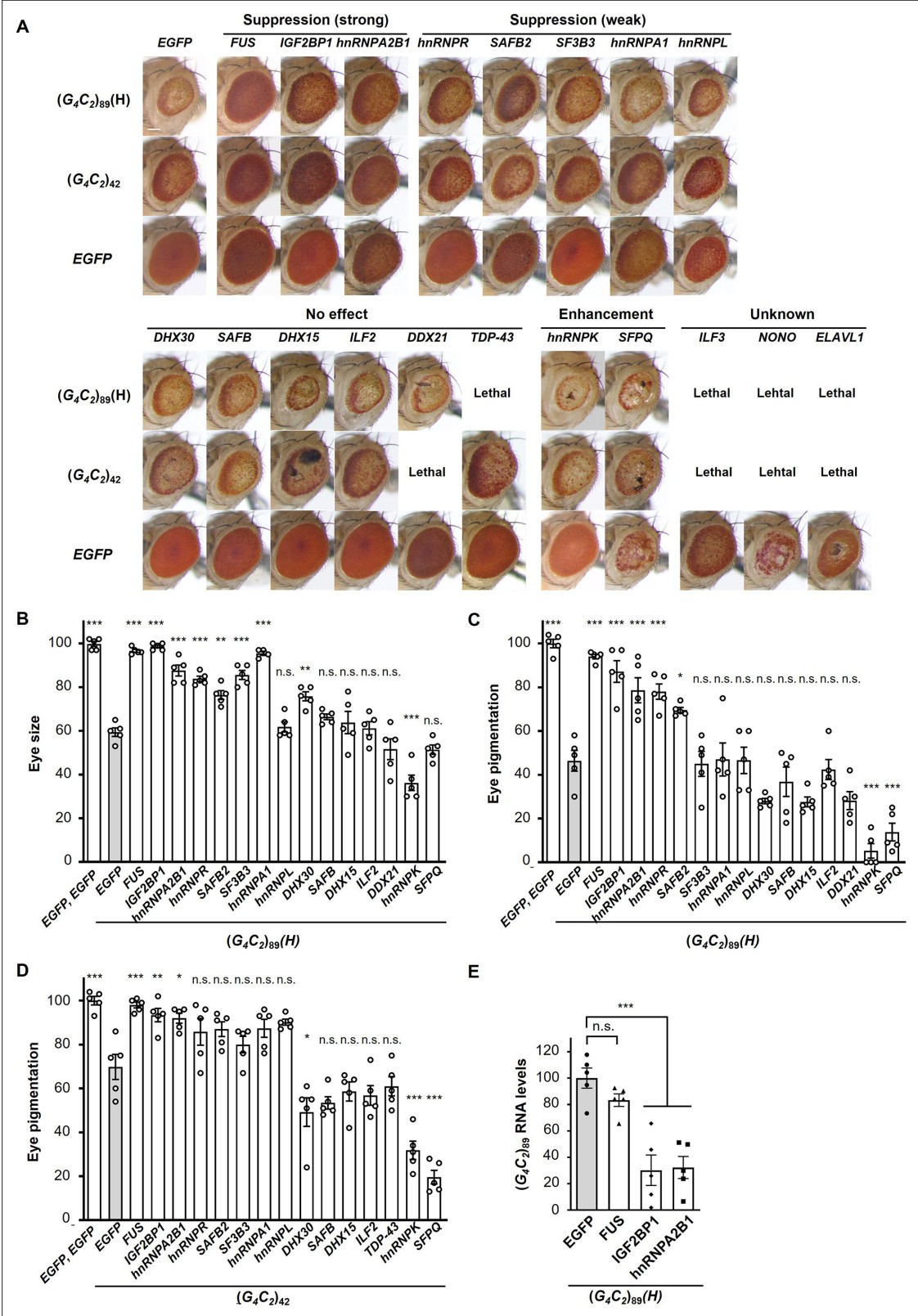

**Figure 1.** Screening for RNA-binding proteins (RBPs) that suppress $G_4C_2$ repeat-induced toxicity in C9-ALS/FTD flies. (**A**) Light microscopic images of the eyes in flies expressing both $(G_4C_2)_{42 \text{ or } 89}$ and the indicated RBPs using the *GMR-Gal4* driver. Coexpression of *FUS*, *IGF2BP1*, or *hnRNPA2B* suppressed eye degeneration in both $(G_4C_2)_{42}$ and $(G_4C_2)_{89}$ flies, indicated by 'Suppression (strong).' Coexpression of *hnRNPR*, *SAFB2*, *SF3B3*, *hnRNPA1*, or *hnRNPL* suppressed eye degeneration in either $(G_4C_2)_{42}$ or $(G_4C_2)_{89}$ flies, indicated by 'Suppression (weak)' (see also **Figure 1—source data 2**).

*Figure 1 continued on next page*

*Figure 1 continued*

Scale bar: 100 μm. (**B**) Quantification of eye size in $(G_4C_2)_{89}$ flies coexpressing the indicated RBPs (n = 5). (**C, D**) Quantification of eye pigmentation in $(G_4C_2)_{89}$ flies (**C**) or $(G_4C_2)_{42}$ flies (**D**) coexpressing the indicated RBPs (n = 5). (**E**) Expression levels of $(G_4C_2)_{89}$ RNA in flies expressing both $(G_4C_2)_{89}$ and the indicated RBPs using the *GMR-Gal4* driver (five independent experiments, n = 25 flies per genotype). The $(G_4C_2)_{89}$(H) fly line expresses $(G_4C_2)_{89}$ RNA at a high level (see also *Figure 1—figure supplement 1*). In (**B–E**), data are presented as the mean ± SEM; p<0.0001, as assessed by one-way ANOVA; n.s., not significant, *p<0.05, **p<0.01, and ***p<0.001, as assessed by Tukey's post hoc analysis. The detailed statistical information is summarized in *Figure 1—source data 3*.

The online version of this article includes the following source data and figure supplement(s) for figure 1:

**Source data 1.** RNA-binding proteins and their cDNA accession numbers screened in the genetic analyses in *Figure 1*.

**Source data 2.** Summary of the genetic analyses in *Figure 1*.

**Source data 3.** Statistical data related to *Figure 1B–E*.

**Figure supplement 1.** Characterization of C9-ALS/FTD flies.

**Figure supplement 1—source data 1.** The artificial sequence inserted in the pUAST vector for generation of $(G_4C_2)_n$ flies.

**Figure supplement 1—source data 2.** Statistical data related to *Figure 1—figure supplement 1C and D*.

**Figure supplement 2.** Coexpression of FUS suppresses $G_4C_2$ repeat-induced toxicity in flies expressing $(G_4C_2)_{89}$.

**Figure supplement 2—source data 1.** Statistical data related to *Figure 1—figure supplement 2B–D*.

IGF2BP1 and hnRNPA2B1 could simply be explained by the decreased levels of $G_4C_2$ repeat RNA, the molecular mechanisms by which FUS suppresses $G_4C_2$ repeat-induced toxicity remain to be clarified. The suppressive effects of FUS on $G_4C_2$ repeat-induced toxicity were confirmed using multiple FUS fly lines, showing the significant suppression of decreased eye size and loss of pigmentation in $(G_4C_2)_{42}$ or $_{89}$ flies coexpressing *FUS* (*Figure 1—figure supplement 2*). Therefore, we decided to further focus on FUS, which is another ALS/FTD-linked RBP, and investigated its mechanism of the suppression of $G_4C_2$ repeat-induced toxicity.

## FUS suppresses $G_4C_2$ repeat-induced toxicity via its RNA-binding activity

We next investigated whether the suppressive effects of FUS on $G_4C_2$ repeat-induced toxicity are mediated by its binding to $G_4C_2$ repeat RNA, using flies expressing FUS with mutations in the RRM (*FUS-RRMmut*), which have been reported to eliminate its RNA-binding activity (*Daigle et al., 2013*). Western blot analysis confirmed that the *FUS-RRMmut* fly line expresses almost an equivalent level of the FUS proteins to the *FUS* fly line (*Figure 2—figure supplement 1*). We found that coexpression of *FUS-RRMmut* did not restore the eye degeneration in flies expressing $(G_4C_2)_{89}$, suggesting that the RNA-binding activity of FUS is essential for its suppressive effects on $G_4C_2$ repeat-induced toxicity (*Figure 2A–C*). We also evaluated the reduced egg-to-adult viability of $(G_4C_2)_{42}$ flies and confirmed that this phenotype was rescued by coexpression of *FUS*, but not by coexpression of *FUS-RRMmut* (*Figure 2D*). Expression of $G_4C_2$ repeat RNA in the nervous system of flies after eclosion using the *elav-GeneSwitch* driver induces motor dysfunction, and coexpression of *FUS* significantly alleviated this motor dysfunction (*Figure 2E*), indicating that FUS suppresses the neuronal phenotypes of flies expressing $G_4C_2$ repeat RNA. It is notable that the motor dysfunction caused by the expression of *FUS* alone was also alleviated by coexpression of $(G_4C_2)_{42}$ (*Figure 2E*), indicating that the $G_4C_2$ repeat RNA conversely suppresses FUS toxicity. This result is consistent with our previous observations in SCA31 flies that UGGAA repeat RNA reduced the aggregation and toxicity of TDP-43 (*Ishiguro et al., 2017*). Moreover, recent studies demonstrated that RNA buffers the phase separation of TDP-43 and FUS, resulting in the suppression of their aggregation (*Maharana et al., 2018*; *Mann et al., 2019*). These findings hence suggest that balancing the crosstalk between repeat RNAs and RBPs neutralizes the toxicities of each other.

## FUS suppresses RNA foci formation and RAN translation from $G_4C_2$ repeat RNA

We next analyzed the effects of FUS expression on RNA foci and DPR production in flies expressing $G_4C_2$ repeat RNA. We performed RNA fluorescence in situ hybridization (FISH) of the salivary glands of fly larvae expressing $(G_4C_2)_{89}$ and found that coexpression of *FUS* significantly decreased the number

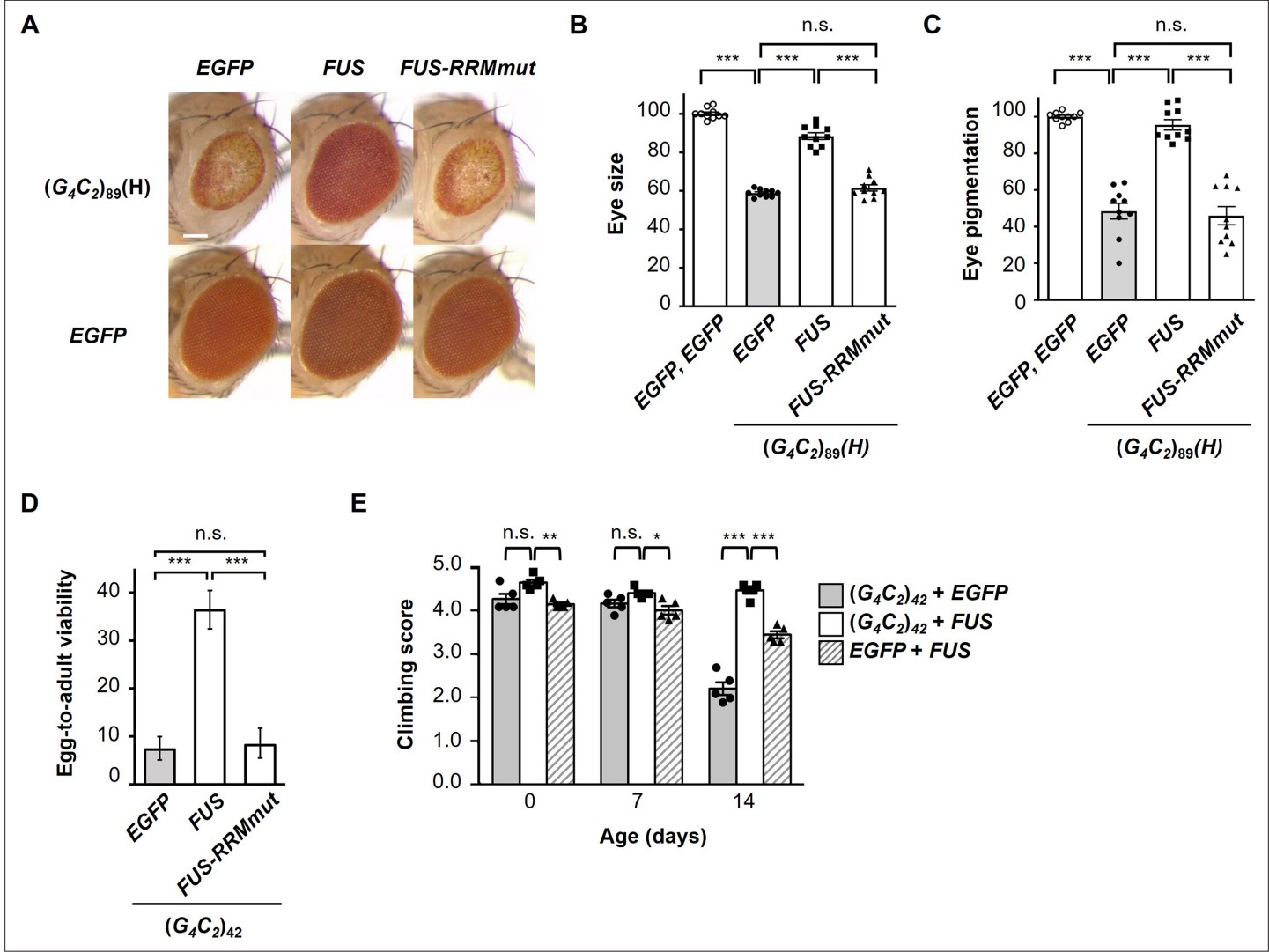

**Figure 2.** FUS suppresses $G_4C_2$ repeat-induced toxicity via its RNA-binding activity. (**A**) Light microscopic images of the eyes in flies expressing both ($G_4C_2)_{89}$ and either *FUS* or *FUS-RRMmut* using the *GMR-Gal4* driver. Scale bar: 100 μm. (**B**) Quantification of eye size in the flies of the indicated genotypes (n = 10). (**C**) Quantification of eye pigmentation in the flies of the indicated genotypes (n = 10). (**D**) Egg-to-adult viability in flies expressing both ($G_4C_2)_{42}$ and either *FUS* or *FUS-RRMmut* using the *GMR-Gal4* driver (>500 flies per genotype). (**E**) Climbing ability in flies expressing both ($G_4C_2)_{42}$ and *FUS* using the *elav-GeneSwitch* driver (five independent experiments, n = 100 flies per each genotype). In (**B–E**), data are presented as the mean ± SEM. In (**B, C**), $p < 0.0001$, as assessed by one-way ANOVA; n.s., not significant, and ***$p < 0.001$, as assessed by Tukey's post hoc analysis. In (**D**), n.s., not significant and ***$p < 0.001$, as assessed by Tukey's multiple-comparison test using wholly significant difference. In (**E**), n.s., not significant, *$p < 0.05$, **$p < 0.01$, and ***$p < 0.001$, as assessed by two-way repeated-measures ANOVA with Tukey's post hoc analysis. The detailed statistical information is summarized in *Figure 2—source data 1*.

The online version of this article includes the following source data and figure supplement(s) for figure 2:

**Source data 1.** Statistical data related to *Figure 2B–E*.

**Figure supplement 1.** Western blot analysis showing expression levels of FUS and FUS-RRMmut proteins.

**Figure supplement 1—source data 1.** Statistical data related to *Figure 2—figure supplement 1B*.

**Figure supplement 1—source data 2.** Source data related to *Figure 2—figure supplement 1A*.

of nuclei containing RNA foci in ($G_4C_2)_{89}$ flies, whereas it was not altered by coexpression of *FUS-RRMmut* (**Figure 3A and B**). We confirmed that the expression levels of $G_4C_2$ repeat RNA in ($G_4C_2)_{89}$ flies were not altered by coexpression of *FUS* or *FUS-RRMmut* (**Figure 3C**). These results were in good agreement with our previous study on SCA31 showing the suppressive effects of FUS and other RBPs on RNA foci formation of UGGAA repeat RNA through altering RNA structures and preventing

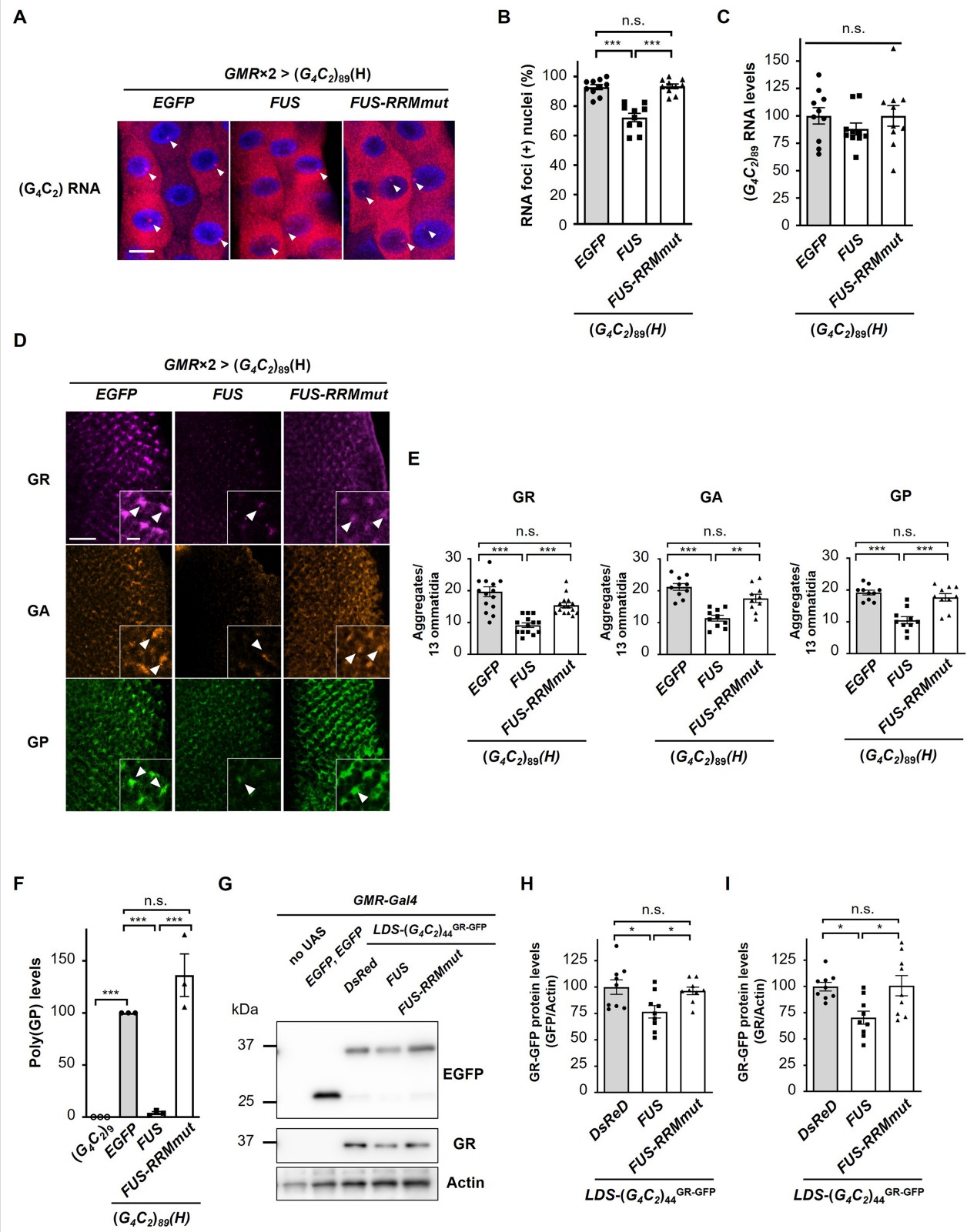

**Figure 3.** FUS suppresses RNA foci formation and RAN translation from $G_4C_2$ repeat RNA. (**A**) Fluorescence in situ hybridization (FISH) analyses of $G_4C_2$ repeat RNA in the salivary glands of fly larvae expressing both $(G_4C_2)_{89}$ and either *FUS* or *FUS-RRMmut* using two copies of the *GMR-Gal4* driver (red: $G_4C_2$ RNA; blue [DAPI]: nuclei). Arrowheads indicate RNA foci. Scale bar: 20 μm. (**B**) Quantification of the number of nuclei containing RNA foci from the FISH analyses in (**A**) (n = 10). (**C**) Expression levels of $(G_4C_2)_{89}$ RNA in fly larvae expressing both $(G_4C_2)_{89}$ and either *FUS* or *FUS-RRMmut* using

*Figure 3 continued on next page*

*Figure 3 continued*

the *GMR-Gal4* driver (10 independent experiments, n = 50 flies per each genotype). (**D**) Immunohistochemical analyses of dipeptide repeat proteins (DPRs) stained with anti-DPR antibodies in the eye imaginal discs of fly larvae expressing both $(G_4C_2)_{89}$ and either *FUS* or *FUS-RRMmut* using two copies of the *GMR-Gal4* driver (magenta: poly(GR); orange: poly(GA); green: poly(GP)). Arrowheads indicate cytoplasmic aggregates. Scale bars: 20 µm (low magnification) or 5 µm (high magnification). (**E**) Quantification of the number of DPR aggregates from the immunohistochemical analyses in (**D**) (n = 14 or 15 [GR], or 10 [GA or GP]). (**F**) Immunoassay to determine poly(GP) levels in flies expressing both $(G_4C_2)_{89}$ and either *FUS* or *FUS-RRMmut* using the *GMR-Gal4* driver (three independent experiments, n = 30 flies per each genotype). (**G**) Western blot analysis of the heads of adult flies expressing both LDS-$(G_4C_2)_{44}$GR-GFP and any of *DsRed*, *FUS* or *FUS-RRMmut* using the *GMR-Gal4* driver, using either an anti-GFP (upper panel) or anti-GR antibody (middle panel). (**H, I**) Quantification of GR-GFP protein levels from the western blot analysis in (**G**) (nine independent experiments, n = 90 flies per each genotype). In (**B**, **C**, **E**, **F**, **H**, **I**), data are presented as the mean ± SEM. In (**B**, **E**, **F**), p<0.0001, as assessed by one-way ANOVA; n.s., not significant, *p<0.05, **p<0.01, and ***p<0.001, as assessed by Tukey's post hoc analysis. In (**C**), p=0.452, as assessed by one-way ANOVA; n.s., not significant, as assessed by Tukey's post hoc analysis. In (**H**), p=0.0148, as assessed by one-way ANOVA; n.s., not significant and *p<0.05, as assessed by Tukey's post hoc analysis. In (**I**), p=0.0072, as assessed by one-way ANOVA; n.s., not significant and *p<0.05, as assessed by Tukey's post hoc analysis. The detailed statistical information is summarized in *Figure 3—source data 1*.

The online version of this article includes the following source data and figure supplement(s) for figure 3:

**Source data 1.** Statistical data related to *Figure 3B, C, E, F, H and I*.

**Source data 2.** Source data related to *Figure 3G*.

**Figure supplement 1.** Schema of the *LDS-$(G_4C_2)_{44}$*GR-GFP construct.

**Figure supplement 2.** Overexpression of *FUS* does not suppress eye degeneration in dipeptide repeat protein (DPR)-only flies expressing DPRs translated from non-$G_4C_2$ RNAs.

aggregation of misfolded repeat RNA as RNA chaperones (*Ishiguro et al., 2017*), raising the possibility that FUS has RNA-chaperoning activity also for $G_4C_2$ repeat RNA. Immunohistochemistry of the eye imaginal discs of fly larvae expressing $(G_4C_2)_{89}$ revealed that coexpression of *FUS* significantly decreased the number of DPR aggregates in $(G_4C_2)_{89}$ flies, whereas coexpression of *FUS-RRMmut* did not (*Figure 3D and E*). Quantitative analyses of poly(GP) by immunoassay also demonstrated that poly(GP) levels were greatly decreased in $(G_4C_2)_{89}$ flies upon coexpression of *FUS*, but not *FUS-RRMmut* (*Figure 3F*), indicating that FUS suppresses RAN translation from the $G_4C_2$ repeat RNA to reduce DPR production. Considering that the 5' upstream sequence of the $G_4C_2$ repeat in the *C9orf72* gene is reported to affect RAN translation activity (*Green et al., 2017*; *Tabet et al., 2018*), we used flies expressing the $G_4C_2$ repeat sequence with the upstream intronic sequence of the *C9orf72* gene, namely, *LDS-$(G_4C_2)_{44}$*GR-GFP (*Goodman et al., 2019*). Since this construct has a 3'-green fluorescent protein (GFP) tag in the GR reading frame downstream of the $G_4C_2$ repeat sequence, the GR-GFP fusion protein is produced by RAN translation (*Figure 3—figure supplement 1*). We confirmed that coexpression of *FUS* significantly decreased the expression level of GR-GFP, whereas coexpression of *FUS-RRMmut* had no effect (*Figure 3G–I*).

We further excluded the possibility that FUS directly interacts with DPRs, rather than with $G_4C_2$ repeat RNA, to decrease DPR levels and exert its suppressive effects. Using DPR-only flies expressing DPRs translated from non-$G_4C_2$ RNAs with alternative codons (*Mizielinska et al., 2014*), we confirmed that FUS did not suppress the eye degeneration in DPR-only flies expressing poly(GR), but rather enhanced their phenotypes, likely due to the additive effects of FUS toxicity (*Figure 3—figure supplement 2*). Together with the finding that FUS decreases not only DPR expression but also RNA foci formation (*Figure 3A and B*), these results collectively indicate that FUS indeed interacts with $G_4C_2$ repeat RNA and regulates RAN translation from $G_4C_2$ repeat RNA in *Drosophila* models of C9-ALS/FTD.

## Reduction of endogenous *caz* expression enhances $G_4C_2$ repeat-induced toxicity, RNA foci formation, and DPR aggregation

To elucidate the physiological role of FUS on RAN translation, we also investigated the role of endogenous *caz*, a *Drosophila* homologue of *FUS*, on $G_4C_2$ repeat-induced toxicity in flies expressing $G_4C_2$ repeat RNAs. Coexpression of *caz* as well as *FUS* suppressed eye degeneration in flies expressing $(G_4C_2)_{42 \text{ or } 89}$ (*Figure 4—figure supplement 1*). These data suggest that *caz* is a functional homologue of *FUS*. In contrast, knockdown of *caz* by RNA interference or its hemizygous deletion modestly but significantly enhanced the eye degeneration in $(G_4C_2)_{89}$ flies (*Figure 4A–D*), indicating that reduced *caz* expression enhances $G_4C_2$ repeat-induced toxicity. We next analyzed the effects of *caz* knockdown

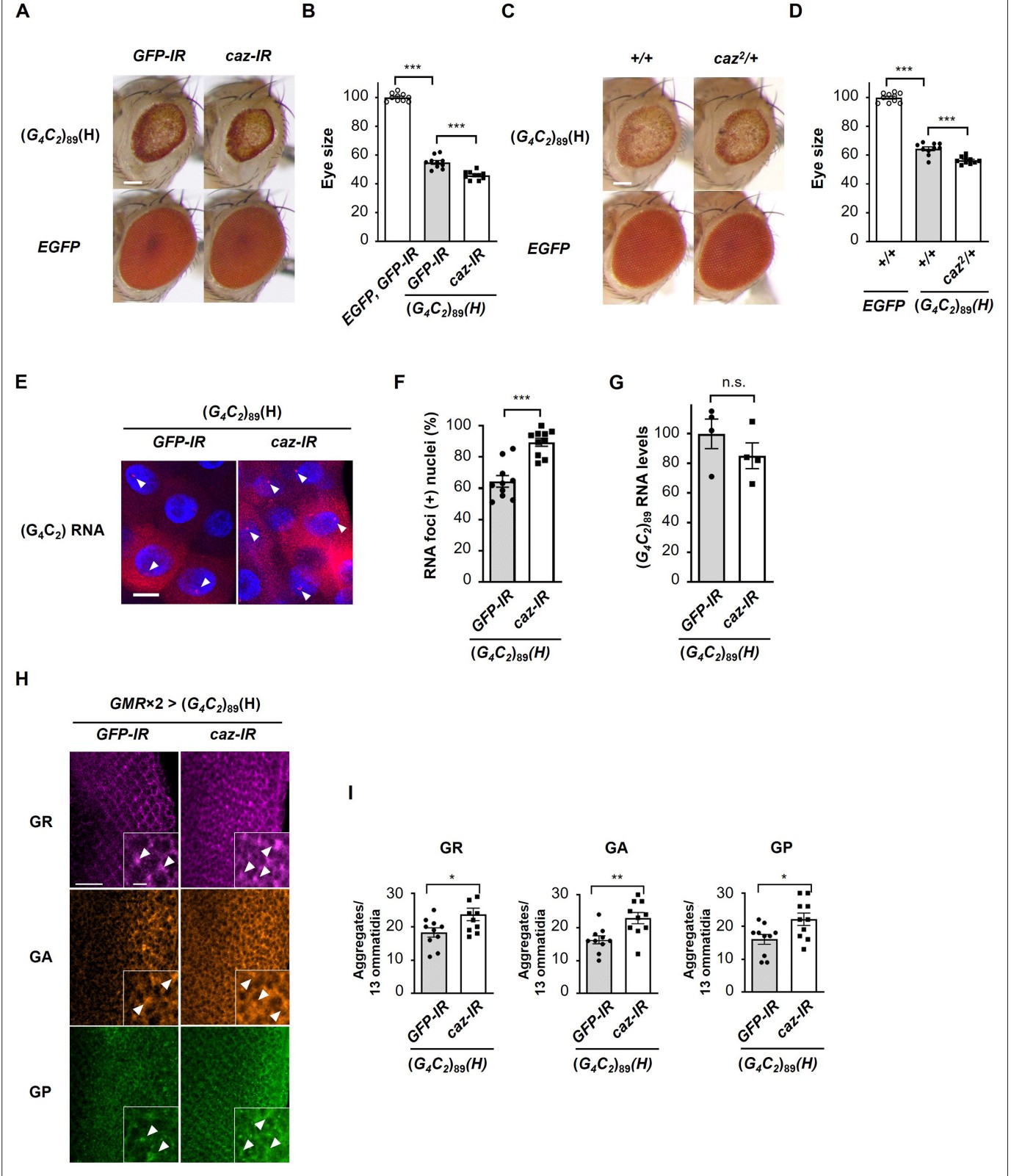

**Figure 4.** Reduction of endogenous *caz* expression enhances $G_4C_2$ repeat-induced toxicity, RNA foci formation, and dipeptide repeat protein (DPR) aggregation. (**A**) Light microscopic images of the eyes in flies expressing $(G_4C_2)_{89}$ using the *GMR-Gal4* driver, with knockdown of *caz*. Scale bar: 100 μm. (**B**) Quantification of eye size in flies of the indicated genotypes shown in (**A**) (n = 10). (**C**) Light microscopic images of the eyes in flies expressing $(G_4C_2)_{89}$ using the *GMR-Gal4* driver, with a hemizygous deletion of *caz*. Scale bar: 100 μm. (**D**) Quantification of eye size in the flies of the indicated genotypes

*Figure 4 continued on next page*

*Figure 4 continued*

shown in (**C**) (n = 10). (**E**) Fluorescence in situ hybridization (FISH) analyses of $G_4C_2$ repeat RNA in the salivary glands of fly larvae expressing $(G_4C_2)_{89}$ using the *GMR-Gal4* driver, with knockdown of *caz* (red: $G_4C_2$ RNA; blue [DAPI]: nuclei). Arrowheads indicate RNA foci. Scale bar: 20 µm. (**F**) Quantification of the number of nuclei containing RNA foci from the FISH analyses in (**E**) (n = 10). (**G**) Expression levels of $(G_4C_2)_{89}$ RNA in fly larvae expressing $(G_4C_2)_{89}$ using the *GMR-Gal4* driver, with knockdown of *caz* (four independent experiments, n = 20 flies per each genotype). (**H**) Immunohistochemical analyses of DPRs stained with anti-DPR antibodies in the eye imaginal discs of fly larvae expressing $(G_4C_2)_{89}$ using two copies of the *GMR-Gal4* driver, with the knockdown of *caz*. (magenta: poly(GR); orange: poly(GA); green: poly(GP)). Arrowheads indicate cytoplasmic aggregates. Scale bars: 20 µm (low magnification) or 5 µm (high magnification). (**I**) Quantification of the number of DPR aggregates from the immunohistochemical analyses in (**H**) (n = 10). In (**B**, **D**, **F**, **G**, **I**), data are presented as the mean ± SEM. In (**B**, **D**), p<0.0001, as assessed by one-way ANOVA; ***p<0.001, as assessed by Tukey's post hoc analysis. In (**F**, **G**, **I**), n.s., not significant, *p<0.05, **p< 0.01, and ***p<0.001, as assessed by the unpaired *t*-test. The detailed statistical information is summarized in *Figure 4—source data 1*.

The online version of this article includes the following source data and figure supplement(s) for figure 4:

**Source data 1.** Statistical data related to *Figure 4B, D, F, G and I*.

**Figure supplement 1.** Endogenous *caz* is a functional homologue of *FUS* for the suppression of $G_4C_2$ repeat-induced toxicity.

**Figure supplement 1—source data 1.** Statistical data related to *Figure 4—figure supplement 1B–D*.

on RNA foci formation and DPR production in flies expressing $(G_4C_2)_{89}$. FISH analysis of the salivary glands revealed that knockdown of *caz* significantly increased the number of nuclei containing RNA foci in $(G_4C_2)_{89}$ flies (*Figure 4E and F*). We also confirmed that the expression levels of $G_4C_2$ repeat RNA in $(G_4C_2)_{89}$ flies were not altered by the knockdown of *caz* (*Figure 4G*). Immunohistochemical analysis showed that knockdown of *caz* significantly increased the number of DPR aggregates in $(G_4C_2)_{89}$ flies (*Figure 4H and I*). These results indicate that the reduction of *caz* expression enhances RNA foci formation and DPR aggregation, compatible with the results of *FUS* coexpression in flies expressing $(G_4C_2)_{89}$ (*Figure 3*), and that FUS functions as an endogenous regulator of RAN translation.

## FUS directly binds to and modulates the G-quadruplex structure of $G_4C_2$ repeat RNA, resulting in the suppression of RAN translation in vitro

We next confirmed the direct interaction of FUS with $G_4C_2$ repeat RNA by the filter binding assay. We found that His-tagged FUS binds to the $(G_4C_2)_4$ RNA in a dose-dependent manner, but not to the control $(AAAAAA)_4$ RNA (*Figure 5A*), and His-tagged FUS-RRMmut had almost no binding affinity to the $(G_4C_2)_4$ RNA, consistent with a previous study (*Mori et al., 2013a*). We also confirmed the interaction of FUS with the $G_4C_2$ repeat RNA in our C9-ALS/FTD flies by showing the colocalization of FUS with the RNA foci (*Figure 5—figure supplement 1*), consistent with a recent study using C9-ALS/FTD patient fibroblasts (*Bajc Česnik et al., 2019*). Since $G_4C_2$ repeat RNA was reported to form both G-quadruplex and hairpin structures (*Fratta et al., 2012*; *Haeusler et al., 2014*; *Reddy et al., 2013*; *Su et al., 2014*), we next characterized the interactions of FUS with $G_4C_2$ repeat RNA having different structures. $G_4C_2$ repeat RNA is known to form G-quadruplex structures in the presence of $K^+$, whereas they form hairpin structures in the presence of $Na^+$ (*Su et al., 2014*). Surface plasmon resonance (SPR) analyses demonstrated that FUS preferentially binds to $(G_4C_2)_4$ RNA with the G-quadruplex structure in KCl buffer (*Table 1*, dissociation constant $(K_D) = 1.5 \times 10^{-8}$ M) and weakly to $(G_4C_2)_4$ RNA with the hairpin structure in NaCl buffer (*Table 1*, $K_D = 1.3 \times 10^{-7}$ µM). We also confirmed that FUS has poor binding affinity to $(G_4C_2)_4$ RNA in LiCl buffer (*Table 1*, $K_D = 1.4 \times 10^{-5}$ µM), which destabilizes the G-quadruplex structure (*Hardin et al., 1992*), and was an almost similar level to its binding affinity to the negative control $(A_4C_2)_4$ RNA (not shown). These results suggest the preferential binding of FUS to $G_4C_2$ repeat RNA with the G-quadruplex structure, which is consistent with a previous report showing preferential binding of FUS to G-quadruplex structured Sc1 and DNMT RNAs (*Ozdilek et al., 2017*). Considering that higher-order structures, including G-quadruplex and hairpin structures, are reported to be involved in RAN translation (*Mori et al., 2021*; *Simone et al., 2018*; *Wang et al., 2019*; *Zu et al., 2011*), we next investigated the effects of FUS on the structure of $G_4C_2$ repeat RNA. The circular dichroism (CD) spectrum of $(G_4C_2)_4$ RNA in KCl buffer was found to exhibit a positive peak at approximately 260 nm and a negative peak at 240 nm (*Figure 5B*, black line), consistent with previous reports (*Fratta et al., 2012*; *Haeusler et al., 2014*; *Reddy et al., 2013*; *Su et al., 2014*). Interestingly, upon the addition of FUS, these two peaks were notably shifted to longer wavelengths with substantial CD

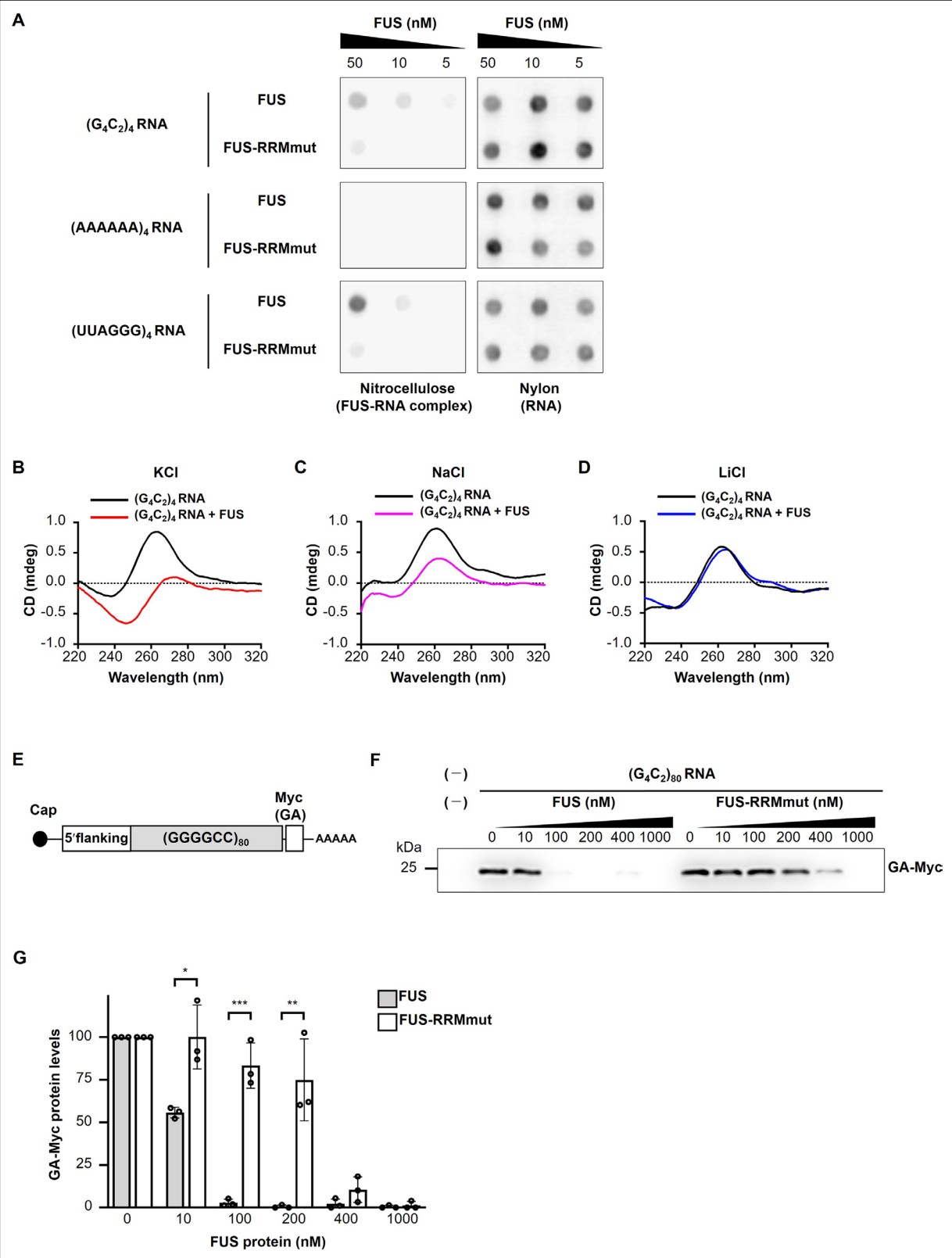

**Figure 5.** FUS directly binds to and modulates the G-quadruplex structure of $G_4C_2$ repeat RNA, resulting in the suppression of RAN translation in vitro. (**A**) Analysis of the binding of His-tagged FUS proteins to biotinylated $(G_4C_2)_4$ RNA by the filter binding assay. The nitrocellulose membrane (left) traps RNA-bound FUS proteins, whereas unbound RNAs are recovered on the nylon membrane (right), and then the RNAs trapped on each of the membranes was probed with streptavidin-horseradish peroxidase (HRP). Biotinylated $(AAAAAA)_4$ and $(UUAGGG)_4$ were used as negative and positive

*Figure 5 continued on next page*

*Figure 5 continued*

controls, respectively. (**B–D**) CD spectra of $(G_4C_2)_4$ RNA incubated with or without FUS in the presence of 150 mM KCl (**B**), NaCl (**C**), or LiCl (**D**). The CD spectrum of FUS alone was subtracted from that of $(G_4C_2)_4$ RNA incubated with FUS. The original data are shown in *Figure 5—figure supplement 2B–2D*. (**E**) Schema of the template RNA containing the $(G_4C_2)_{80}$ sequence and 113 nucleotides of the 5'-flanking region of intron 1 of the human *C9orf72* $G_4C_2$ repeat sequence. A Myc tag in the GA frame was introduced downstream of the $(G_4C_2)_{80}$ repeat sequence. (**F**) Western blot analysis of samples from in vitro translation using rabbit reticulocyte lysate in the presence or absence of increasing concentrations of FUS or FUS-RRMmut. The GA-Myc fusion protein was detected by western blotting using the anti-Myc antibody. (**G**) Quantification of the GA-Myc fusion protein in (**F**) (n = 3). In (**G**), data are presented as the mean ± SEM; *p<0.05, **p<0.01, and ***p<0.001, as assessed by the unpaired *t*-test. The detailed statistical information is summarized in *Figure 5—source data 1*.

The online version of this article includes the following source data and figure supplement(s) for figure 5:

**Source data 1.** Statistical data related to *Figure 5G*.

**Source data 2.** Source data related to *Figure 5F*.

**Figure supplement 1.** FUS colocalizes with $G_4C_2$ RNA foci.

**Figure supplement 2.** FUS modulates the G-quadruplex structure of $G_4C_2$ repeat RNA.

spectrum changes, indicating a significant structural alteration in $(G_4C_2)_4$ RNA (*Figure 5B*, red line). We confirmed that the CD spectrum of FUS alone in the wavelength range of 240–300 nm was almost negligible (*Figure 5—figure supplement 2A*, green line), indicating that this change in CD spectrum is attributed to structural changes in the $(G_4C_2)_4$ RNA. We also observed CD spectrum changes to some extent in the $(G_4C_2)_4$ RNA upon the addition of FUS in NaCl buffer, but not in LiCl buffer, confirming an interaction between FUS and hairpin-structured $(G_4C_2)_4$ RNA as well (*Figure 5C and D*). We further analyzed the interaction between FUS and $G_4C_2$ repeat RNA by imino proton nuclear magnetic resonance (NMR). In KCl buffer, the NMR signals of the imino proton for the G-quadruplex structure of $(G_4C_2)_4$ RNA were detected in the region around 10–12 ppm (*Figure 5—figure supplement 2D*), consistent with previous studies (*Fratta et al., 2012*; *Su et al., 2014*). Upon the addition of FUS, the NMR intensities of $(G_4C_2)_4$ RNA were decreased in an FUS concentration-dependent manner (*Figure 5—figure supplement 2D*), further supporting the possibility that that FUS interacts with and modulates the G-quadruplex structure of $(G_4C_2)_4$ RNA. Collectively, these results indicate that FUS directly binds to $G_4C_2$ repeat RNA, preferentially to its G-quadruplex form and modulates its higher-order structures. These structural alterations of $G_4C_2$ repeat RNAs by FUS did not require ATP or interactions with other proteins, suggesting its role as an RNA chaperone for $G_4C_2$ repeat RNA (*Rajkowitsch et al., 2007*).

To further clarify the direct link between the binding of FUS to $G_4C_2$ repeat RNA and its effects on RAN translation, we employed a cell-free in vitro translation assay using rabbit reticulocyte lysate. We designed a reporter construct containing the 80 $G_4C_2$ repeat sequence with the 5' upstream intronic sequence of the *C9orf72* gene and the Myc tag sequence in the GA reading frame at the 3' downstream (*Figure 5E*). This upstream sequence contained multiple stop codons in each reading frame and lacked AUG initiation codons. We confirmed by western blotting that this reporter system indeed produces GA-Myc by RAN translation, consistent with previous studies (*Green et al., 2017*; *Tabet et al., 2018*). Notably, upon the addition to this translation system, FUS suppressed RAN translation efficiently, whereas FUS-RRMmut did not. FUS decreased the expression levels of GA-Myc at as low as 10 nM and nearly eliminated RAN translation activity at 100 nM. At 400 nM, FUS-RRMmut weakly suppressed the GA-Myc expression levels probably because of the residual RNA-binding activity (*Figure 5F and G*). Taken together, these results indicate that FUS suppresses RAN translation in vitro through direct interactions with $G_4C_2$ repeat RNA as an RNA chaperone.

**Table 1.** Association ($k_a$) and dissociation ($k_d$) rate and dissociation constants ($K_D$) between FUS and $(G_4C_2)_4$ RNA in different buffers as assessed by surface plasmon resonance (SPR) analysis.

| Buffer | $k_a$ (M$^{-1}$s$^{-1}$) × 10$^6$ | $k_d$ (s$^{-1}$) × 10$^{-3}$ | $K_D$ (M) |
|---|---|---|---|
| KCl | 1.4 | 22 | $1.5 \times 10^{-8}$ |
| NaCl | 0.41 | 54 | $1.3 \times 10^{-7}$ |
| LiCl | 0.0018 | 25 | $1.4 \times 10^{-5}$ |

# Identification of G-quadruplex-targeting RBPs that suppress $G_4C_2$ repeat-induced toxicity in C9-ALS/FTD flies

Considering that FUS suppresses $G_4C_2$ repeat-induced toxicity as an RNA chaperone through its preferential binding to the G-quadruplex structure of $G_4C_2$ repeat RNA (*Figure 5* and *Table 1*), we hypothesized that other G-quadruplex-targeting RBPs might have similar suppressive effects on $G_4C_2$ repeat-induced toxicity. To investigate this possibility, we selected six representative G-quadruplex-targeting RBPs, all of which are known to bind to $G_4C_2$ RNA as well (*Cooper-Knock et al., 2014*; *Haeusler et al., 2014*; *Mori et al., 2013a*; *Xu et al., 2013*; *Figure 6—source data 1*). Intriguingly, coexpression of *EWSR1*, *DDX3X*, *DDX5*, or *DDX17* significantly suppressed eye degeneration in $(G_4C_2)_{89}$ flies without altering $G_4C_2$ RNA expression (*Figure 6A–D*). As expected, these RBPs also decreased the number of poly(GA) aggregates in the eye imaginal discs (*Figure 6E and F*). Their effects on $G_4C_2$ repeat-induced toxicity, repeat RNA expression, and RAN translation were consistent with those of FUS. In support of our results, DDX3X was previously reported to suppress RAN translation and $G_4C_2$ repeat-induced toxicity in cell culture in a helicase-activity-dependent manner (*Cheng et al., 2019*). On the other hand, coexpression of *DHX9* or *DHX36* suppressed eye degeneration by reducing $G_4C_2$ repeat RNA levels (*Figure 6A–D*). Since G-quadruplex-targeting RBPs have diverse biological functions, including transcription, RNA processing, translation, and RNA stabilization (*Dumas et al., 2021*), these different effects among G-quadruplex-targeting RBPs on $G_4C_2$ repeat RNA expression might be attributed to their different roles in RNA metabolism. Thus, some G-quadruplex-targeting RBPs regulate RAN translation and $G_4C_2$ repeat-induced toxicity by binding to and possibly by modulating the G-quadruplex structure of $G_4C_2$ repeat RNA.

## Discussion

In this study, we revealed a novel regulatory mechanism of RAN translation from expanded $G_4C_2$ repeat RNA by the ALS/FTD-linked RBP FUS, which suppresses DPR production and neurodegeneration in C9-ALS/FTD *Drosophila* models (*Figures 1–4*). FUS directly binds to $G_4C_2$ repeat RNA and modulates its G-quadruplex structure as evident by CD and NMR analyses (*Figure 5*, *Figure 5—figure supplement 2*), and suppresses RNA foci formation in vivo (*Figure 3A and B*), suggesting its functional role as an RNA chaperone. This is reminiscent of our recent study on SCA31, in which we demonstrated a novel role of the ALS/FTD-linked RBPs TDP-43, FUS, and hnRNPA2B1 as RNA chaperones binding to UGGAA repeat RNA and altering its structure, resulting in the suppression of its neurotoxicity through reducing RNA foci formation and repeat polypeptide translation (*Ishiguro et al., 2017*). Considering the similarities of the effects of FUS on $G_4C_2$ repeat RNA and UGGAA repeat RNA, we conclude that FUS functions as an RNA chaperone also for $G_4C_2$ repeat RNA to regulate its RAN translation.

The suppressive effects of RBPs in several noncoding repeat expansion diseases by the amelioration of their sequestration into RNA foci have been reported. For example, in myotonic dystrophy type 1, MBNL1 was shown to be sequestered into CUG repeat RNA foci, and overexpression of MBNL1 in a mouse model was found to compensate its functional loss, resulting in the reversal of myotonia (*Kanadia et al., 2006*). Similarly, previous studies reported the suppressive effects of other RBPs on neurodegeneration, such as hnRNPA2B1 in fragile X ataxia/tremor syndrome (*Sofola et al., 2007*), and Pur-α and Zfp106 in C9-ALS/FTD (*Xu et al., 2013*; *Celona et al., 2017*). The suppressive effects of these RBPs have been thought to result from the supplementation against their loss-of-function due to their sequestration into RNA foci, although their effects on gain-of-toxic disease pathomechanisms, that is, RAN translation and repeat RNA expression, remain to be elucidated. In contrast, in this study we demonstrated that FUS suppresses neurodegeneration in C9-ALS/FTD by directly targeting $G_4C_2$ repeat RNA and inhibiting RAN translation as an RNA chaperone. Similar suppressive effects of RBPs by targeting UGGAA repeat RNA in SCA31 as RNA chaperones have also been reported (*Ishiguro et al., 2017*). In addition, we also showed that the expression of IGF2BP1, hnRNPA2B1, DHX9, and DHX36 decreased $G_4C_2$ repeat RNA expression and suppressed eye degeneration in our C9-ALS/FTD *Drosophila* model (*Figures 1 and 6*), likely via the reduction of DPR levels. Similarly, we recently reported that hnRNPA3 reduces $G_4C_2$ repeat RNA expression levels, leading to the suppression of neurodegeneration in C9-ALS/FTD fly models (*Taminato et al., 2023*). Interestingly, these RBPs have been reported to be involved in RNA decay pathways as components of the P-body or interactors with the RNA deadenylation machinery (*Tran et al., 2004*; *Katahira et al., 2008*; *Geissler et al., 2016*;

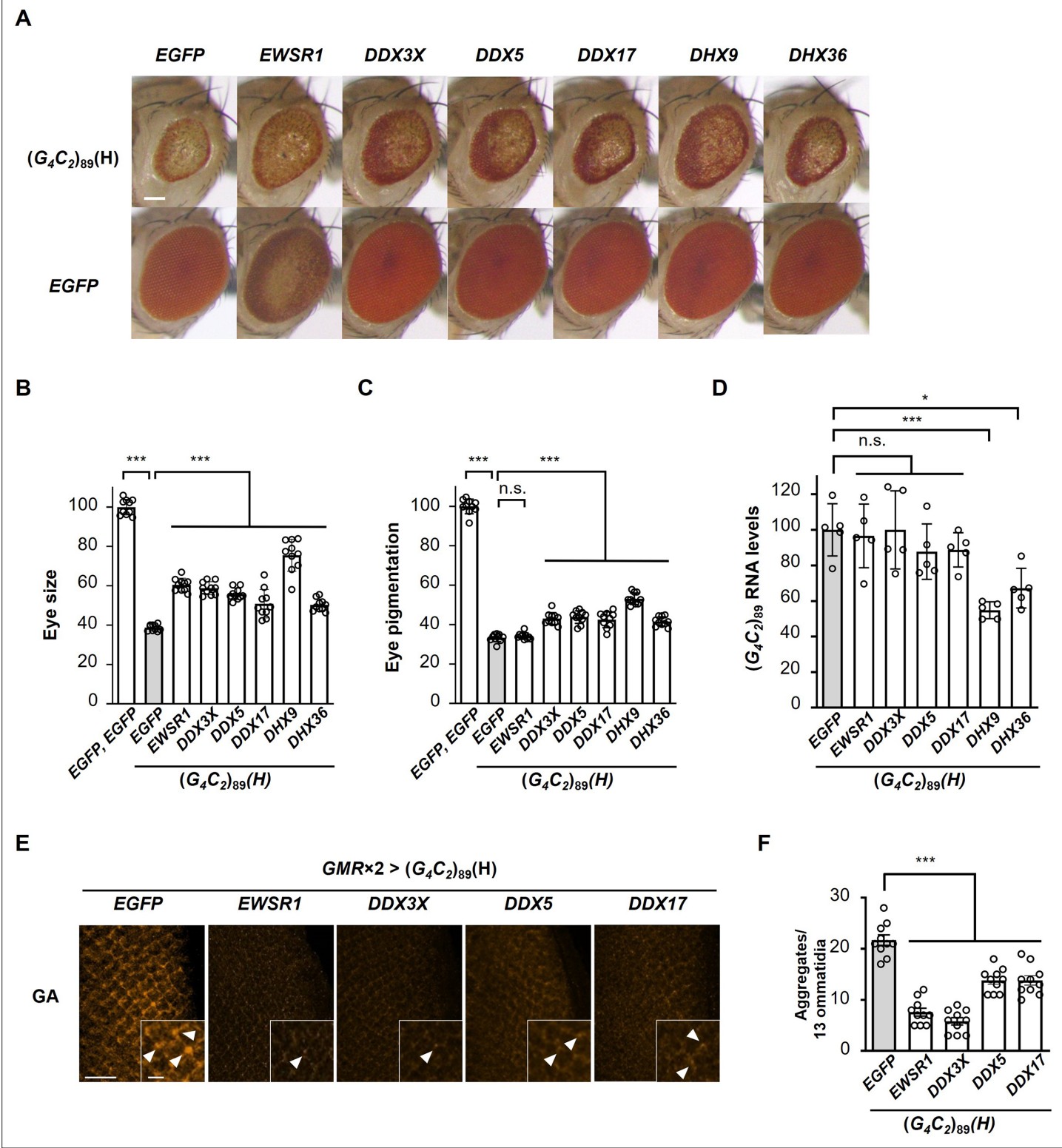

**Figure 6.** Identification of G-quadruplex-targeting RNA-binding proteins (RBPs) that suppress $G_4C_2$ repeat-induced toxicity in C9-ALS/FTD flies.
(**A**) Light microscopic images of eyes in flies expressing both $(G_4C_2)_{89}$ and the indicated G-quadruplex-targeting RBPs using the *GMR-Gal4* driver. Scale bar: 100 µm. (**B**) Quantification of eye size in the flies of the indicated genotypes (n = 10). (**C**) Quantification of eye pigmentation in the flies of the indicated genotypes (n = 10). (**D**) Expression levels of $(G_4C_2)_{89}$ RNA in flies expressing both $(G_4C_2)_{89}$ and the indicated G-quadruplex-targeting RBPs using the *GMR-Gal4* driver (five independent experiments, n = 25 flies per each genotype). (**E**) Immunohistochemical analyses of poly(GA) stained with anti-GA antibody in the eye imaginal discs of fly larvae expressing both $(G_4C_2)_{89}$ and the indicated G-quadruplex-targeting RBPs using two copies of the

*Figure 6 continued on next page*

*Figure 6 continued*

*GMR-Gal4* driver (orange: poly(GA)). Arrowheads indicate cytoplasmic aggregates. Scale bars: 20 μm (low magnification) or 5 μm (high magnification). (**F**) Quantification of the number of poly(GA) aggregates from the immunohistochemical analyses in (**E**) (n = 10). In (**B**, **C**, **D**, **F**), data are presented as the mean ± SEM; p<0.0001, as assessed by one-way ANOVA; n.s., not significant, \*p<0.05 and \*\*\*p<0.001, as assessed by Tukey's post hoc analysis. The detailed statistical information is summarized in *Figure 6—source data 2*.

The online version of this article includes the following source data for figure 6:

**Source data 1.** RNA-binding proteins and their cDNA accession numbers screened in the genetic analyses in *Figure 6*.

**Source data 2.** Statistical data related to *Figure 6B–D, F*.

*Hubstenberger et al., 2017*), possibly contributing to the reduced expression levels of $G_4C_2$ repeat RNA. In myotonic dystrophy type 2 models, MBNL1 was also reported to retain CCUG repeat RNA in the nucleus, resulting in the suppression of RAN translation (*Zu et al., 2017*), implying various mechanisms of the effects of RBPs depending on the combination of RBPs and repeat RNA. Nevertheless, our findings highlighted the previously unrecognized roles of RBPs directly interacting with repeat RNA and modulating gain-of-toxic pathomechanisms, including RAN translation in noncoding repeat expansion diseases.

Several studies have indicated the importance of higher-order structures of repeat RNA in RAN translation. In SCA8 models, hairpin-forming CAG repeat RNA was shown to be RAN-translated to produce polyglutamine proteins, but switching the CAG repeats to non-hairpin-forming CAA repeats abolished the RAN translation (*Zu et al., 2011*). In C9-ALS/FTD, $G_4C_2$ repeat RNA has been reported to form both hairpin and G-quadruplex structures (*Fratta et al., 2012*; *Haeusler et al., 2014*; *Reddy et al., 2013*; *Su et al., 2014*). Although the effect of each structure on RAN translation remains largely unknown, small molecules binding to the hairpin structure or the G-quadruplex structure were both reported to inhibit RAN translation from the $G_4C_2$ repeat RNA, resulting in reduced DPR levels (*Wang et al., 2019*; *Mori et al., 2021*). These findings are in accordance with our results showing that FUS modifies the G-quadruplex structure as well as the hairpin structure of $G_4C_2$ repeat RNA as an RNA chaperone and reduces DPR production. We further found that G-quadruplex-targeting RNA helicases, including DDX3X, DDX5, and DDX17, which are known to bind to $G_4C_2$ repeat RNA (*Cooper-Knock et al., 2014*; *Haeusler et al., 2014*; *Mori et al., 2013a*; *Xu et al., 2013*), also suppress RAN translation and $G_4C_2$ repeat-induced toxicity without altering the expression levels of $G_4C_2$ repeat RNA in our *Drosophila* models. These results suggest that not only ATP-independent RNA chaperones, but also ATP-dependent RNA helicases may regulate RAN translation through modifying the higher-order structures of template repeat RNA. Consistently, a previous study also reported that DDX3X inhibits RAN translation from $G_4C_2$ repeat RNA in a helicase activity-dependent manner (*Cheng et al., 2019*). Knockdown of another RNA helicase, DHX36, has been reported to both promote (*Cheng et al., 2019*) and inhibit (*Liu et al., 2021*; *Tseng et al., 2021*) RAN translation, possibly due to the different effects on repeat RNA structures depending on the experimental conditions. Unfortunately, most of these studies reporting the effects of RBPs on RAN translation have limitations of the detailed structural analyses of repeat RNA. In this study, focusing on FUS, we performed a series of molecular structural analyses, in vitro translation assays, and in vivo genetic analyses to clarify the structure–function relationship of $G_4C_2$ repeat RNA and provide compelling evidence for the modifying effects of FUS on repeat RNA structures leading to the suppression of RAN translation and repeat-induced toxicity in vivo.

FUS has an RRM domain for RNA binding and a low complexity (LC) domain involved in protein interactions, and exerts multifaceted functions, such as RNA transcription, RNA splicing, RNA transport, and formation of membraneless organelles, such as stress granules and nuclear paraspeckles via liquid–liquid phase separation (*Lagier-Tourenne et al., 2010*). Recent studies reported that arginine-rich DPRs, such as poly(GR) and poly(PR), interact with LC domain-containing RBPs, including FUS, and alter their liquid–liquid phase separation, resulting in the disruption of the dynamics and functions of membraneless organelles (*Kwon et al., 2014*; *Lee et al., 2016*; *Lin et al., 2016*). These findings raise the possibility that FUS may exert its suppressive effects by directly interacting with DPRs. However, we showed that FUS does not suppress eye degeneration in DPR-only flies (*Figure 3—figure supplement 2*), indicating that a direct interaction between FUS and DPRs is unlikely to be the mechanism of the suppression of DPR toxicity in our C9-ALS/FTD flies. This result supports our conclusion that FUS suppresses $G_4C_2$ repeat-induced toxicity through direct binding to $G_4C_2$ repeat RNA.

In summary, we here provided evidence that FUS modulates the structure of $G_4C_2$ repeat RNA as an RNA chaperone and regulates RAN translation, resulting in the suppression of neurodegeneration in C9-ALS/FTD fly models. Recent advances in genome sequencing technology unveiled that such expansions of repeat sequences cause more than 50 monogenic human diseases (*Malik et al., 2021*) and are also associated with psychiatric diseases such as autism (*Mitra et al., 2021*; *Trost et al., 2020*). Thus, our findings contribute to the elucidation of the repeat-associated pathogenic mechanisms underlying not only C9-ALS/FTD, but also a broader range of neuromuscular and neuropsychiatric diseases than previously thought, and will advance the development of potential therapies for these diseases.

# Materials and methods

## Key resources table

| Reagent type (species) or resource | Designation | Source or reference | Identifiers | Additional information |
|---|---|---|---|---|
| Strain, strain background (*Drosophila melanogaster*) | UAS-$(G_4C_2)_n$, UAS-FUS-2, UAS-FUS-RRMmut | This paper | N/A | See 'Generation of constructs and transgenic flies' |
| Strain, strain background (*D. melanogaster*) | UAS-RBP (FUS-3; IGF2BP1; hnRNPA2B1; hnRNPR; SAFB2; SF3B3; hnRNPA1; hnRNPL; DHX30; SAFB; DHX15; ILF2; DDX21; hnRNPK; SFPQ; ILF3; NONO; ELAVL1; DDX3X; DDX5; DDX17; DHX9; DHX36) | This paper | N/A | See 'Generation of constructs and transgenic flies' |
| Strain, strain background (*D. melanogaster*) | UAS-LDS-$(G4C2)_{44}^{GR-GFP}$ | *Goodman et al., 2019* (PMID::31110321) | FLYB: FBtp0135960 | |
| Strain, strain background (*D. melanogaster*) | UAS-FUS | *Ishiguro et al., 2017* (PMID::28343865) | FLYB: FBtp0117594 | |
| Strain, strain background (*D. melanogaster*) | UAS-FUS-4 (UAS-FLAG-FUS) | *Wang et al., 2011* (PMID::21881207) | FLYB: FBtp0070284 | |
| Strain, strain background (*D. melanogaster*) | UAS-caz (UAS-FLAG-caz) | *Wang et al., 2011* (PMID:21881207) | FLYB: FBtp0070279 | |
| Strain, strain background (*D. melanogaster*) | $caz^2$ | *Frickenhaus et al., 2015* (PMID::25772687) | FLYB: FBal0323133 | |
| Strain, strain background (*D. melanogaster*) | UAS-TDP-43 | *Ishiguro et al., 2017* (PMID::28343865) | FLYB: FBtp0117592 | |
| Strain, strain background (*D. melanogaster*) | GMR-GAL4 driver | *Yamaguchi et al., 1999* (PMID:10597285) | FLYB: FBtp0010074 | |
| Strain, strain background (*D. melanogaster*) | Elav-GAL4 driver: P{w[+mC]=GAL4-elav.L}2/CyO | Bloomington *Drosophila* Stock Center | BDSC: 8765; FLYB: FBst0008765 | |
| Strain, strain background (*D. melanogaster*) | Elav-GeneSwitch GAL4 driver: y(1) w[*]; P{w[+mC]=elav-Switch.O} GSG301 | Bloomington *Drosophila* Stock Center | BDSC: 43642; FLYB: FBst0043642 | |
| Strain, strain background (*D. melanogaster*) | UAS-EGFP: w[*]; P{w[+mC]=UAS-2xEGFP}AH2 | Bloomington *Drosophila* Stock Center | BDSC: 6874; FLYB: FBst0006874 | |
| Strain, strain background (*D. melanogaster*) | UAS-DsRed: w[*]; P{w[+mC]=UAS-AUG-DsRed}A | Bloomington *Drosophila* Stock Center | BDSC: 6282; FLYB: FBst0006282 | |

*Continued on next page*

*Continued*

| Reagent type (species) or resource | Designation | Source or reference | Identifiers | Additional information |
|---|---|---|---|---|
| Strain, strain background (*D. melanogaster*) | UAS-EWSR1: w[1118]; P{w[+mC]=UAS-EWSR1.C}26M | Bloomington *Drosophila* Stock Center | BDSC: 79592; FLYB: FBst00079592 | |
| Strain, strain background (*D. melanogaster*) | UAS-(GR)$_{36}$: w[1118]; P{{y[+t7.7] w[+mC]=UAS-poly-GR.PO-36} attP40 | Bloomington *Drosophila* Stock Center | BDSC: 58692; FLYB: FBst00058692 | |
| Strain, strain background (*D. melanogaster*) | UAS-(GA)$_{36}$: w[1118]; P{{y[+t7.7] w[+mC]=UAS-poly-GA.PO-36} attP40 | Bloomington *Drosophila* Stock Center | BDSC: 58693; FLYB: FBst00058693 | |
| Strain, strain background (*D. melanogaster*) | UAS-(GR)$_{100}$: w[1118]; P{{y[+t7.7] w[+mC]=UAS-poly-GR.PO-100} attP40 | Bloomington *Drosophila* Stock Center | BDSC: 58696; FLYB: FBst00058696 | |
| Strain, strain background (*D. melanogaster*) | UAS-(GA)$_{100}$: w[1118]; P{{y[+t7.7] w[+mC]=UAS-poly-GA.PO-100} attP40 | Bloomington *Drosophila* Stock Center | BDSC: 58697; FLYB: FBst00058697 | |
| Strain, strain background (*D. melanogaster*) | RNAi of *GFP*: w[1118]; P{w[+mC]=UAS-GFP.dsRNA.R}142 | Bloomington *Drosophila* Stock Center | BDSC: 9330; FLYB: FBst0009330 | |
| Strain, strain background (*D. melanogaster*) | RNAi of *caz*: P{KK107486}VIE-260B | Vienna *Drosophila* Resource Center | VDRC: v100291; FLYB: FBst0472165 | |
| Antibody | Rat monoclonal anti-poly(GR) antibody (5A2) | Millipore | Car# MABN778; RRID:AB_2728664 | IHC(1:1000), WB(1:1000) |
| Antibody | Mouse monoclonal anti-poly(GA) antibody (5E9) | Millipore | Car# MABN889; RRID:AB_2728663 | IHC(1:1000) |
| Antibody | Rabbit polyclonal anti-poly(GA) antibody | Cosmo Bio | Cat# CAC-TIP-C9-P01 | IHC(1:1000) |
| Antibody | Rabbit polyclonal anti-poly(GP) antibody | Novus Biologicals | Cat# NBP2-25018; RRID:AB_2893239 | IHC(1:1000) |
| Antibody | Rabbit polyclonal anti-FUS antibody | Bethyl Laboratories | Cat# A300-302A; RRID:AB_309445 | IHC(1:1000), WB(1:1000) |
| Antibody | Mouse monoclonal anti-EGFP antibody | Clontech | Cat# 632569 | WB(1:1000) |
| Antibody | Mouse monoclonal anti-actin antibody (AC-40) | Sigma-Aldrich | Cat# A4700; RRID:AB_476730 | WB(1:1000) |
| Antibody | Mouse monoclonal anti-c-Myc antibody (9E10) | Wako | Cat# 017-21876 | WB(1:3000) |
| Recombinant DNA reagent | pcDNA5/FRT-*C9orf72* intron1-(G$_4$C$_2$)$_{80}$ (plasmid) | This paper | | See 'RNA synthesis for in vitro translation' |
| Sequence-based reagent | (G$_4$C$_2$)$_n$_F(1) | This paper | PCR primers | ATGAATGGGAG CAGTGGTGG |
| Sequence-based reagent | (G$_4$C$_2$)$_n$_R(1) | This paper | PCR primers | TGTTGAGAGTCA GCAGTAGCC |
| Sequence-based reagent | (G$_4$C$_2$)$_n$_F(2) | This paper | PCR primers | CCCAATCCATATG ACTAGTAGATCC |
| Sequence-based reagent | (G$_4$C$_2$)$_n$_R(2) | This paper | PCR primers | TGTAGGTAGTTTGT CCAATTATGTCA |
| Sequence-based reagent | gal4_F | *Li et al., 2008* (PMID:18449188) | PCR primers | TTGAAATCGC GTCGAAGGA |

*Continued on next page*

*Continued*

| Reagent type (species) or resource | Designation | Source or reference | Identifiers | Additional information |
|---|---|---|---|---|
| Sequence-based reagent | *gal4_R* | *Li et al., 2008* (PMID:18449188) | PCR primers | GGCTCCAATGG CTAATATGCA |
| Peptide, recombinant protein | His-FUS | This paper | N/A | See 'Filter binding assay' |
| Peptide, recombinant protein | His-FUS-RRMmut | This paper | N/A | See 'Filter binding assay' |
| Peptide, recombinant protein | FUS (not tagged) | This paper | N/A | See 'Preparation of recombinant FUS protein' |
| Peptide, recombinant protein | FUS-RRMmut (not tagged) | This paper | N/A | See 'Preparation of recombinant FUS protein' |
| Commercial assay or kit | In-Fusion Cloning system | TaKaRa Bio | Cat# Z9645N | |
| Commercial assay or kit | EZ-Tn5<KAN-2>Insertion Kit | Epicentre | Cat# EZI011RK | |
| Commercial assay or kit | QuantiTect Reverse Transcription Kit | QIAGEN | Cat# 205314 | |
| Commercial assay or kit | mMESSAGE mMACHINE T7 Transcription Kit | Thermo Fisher Scientific | Cat# AM1344 | |
| Commercial assay or kit | Flexi Rabbit Reticulocyte Lysate System | Promega | Cat# L4540 | |
| Chemical compound, drug | RU486 (mifepristone) | Wako | M3321; CAS: 84371-65-3 | |
| Chemical compound, drug | Formula 4-24 Instant *Drosophila* medium | Wako | Cat# 534-20571 | |
| Software, algorithm | ZEN imaging software | Zeiss | RRID:SCR_013672; https://www.zeiss.com/microscopy/en/products/software/zeiss-zen.html | |
| Software, algorithm | ImageJ | *Schneider et al., 2012* (PMID:22930834) | RRID:SCR_003070; https://imagej.nih.gov/ij/ | |
| Software, algorithm | GraphPad Prism version 8.4.3 | GraphPad Software Inc. | RRID:SCR_002798; https://www.graphpad.com | |

## Flies

All fly stocks were cultured and crossed at 23°C or 25°C in standard cornmeal-yeast-glucose medium. Male adult flies were used for the climbing assay and *GeneSwitch* experiments. 3- to 5-day-old female adult flies were used for the evaluation of eye phenotype using a stereoscopic microscope model SZX10 (Olympus). Female third-instar larvae were used for quantitative real-time polymerase chain reaction (PCR), RNA FISH, and immunohistochemistry experiments. The transgenic fly line bearing the *GMR-Gal4* transgene has been described previously (*Yamaguchi et al., 1999*). The transgenic fly lines bearing *elav-Gal4* (#8765), *elav-GeneSwitch* (#43642), *UAS-EGFP* (#6874), *UAS-DsRed* (#6282), *UAS-GFP-IR* (inverted repeat) (#9330), *UAS-(GR)$_{36}$* (#58692), *UAS-(GA)$_{36}$* (#58693), *UAS-(GR)$_{100}$* (#58696), *UAS-(GA)$_{100}$* (#58697), and *UAS-EWSR1* (#79592) were obtained from Bloomington *Drosophila* Stock Center. The transgenic fly line bearing *UAS-caz-IR* (#100291) was obtained from Vienna *Drosophila* Resource Center. The fly line with the *caz* null allele (*caz$^2$*), *UAS-LDS-(G$_4$C$_2$)$_{44}$$^{GR-GFP}$*, and *UAS-caz* (*UAS-FLAG-caz*) and *UAS-FUS*-4 (*UAS-FLAG-FUS*) were kind gifts from Dr. Erik Storkebaum (*Frickenhaus et al., 2015*), Dr. Nancy Bonini (*Goodman et al., 2019*), and Dr. Brian McCabe (*Wang et al., 2011*), respectively. Other transgenic fly lines were generated in this study. Full genotypes of the fly lines used in all figures and their cultured temperatures are described in *Supplementary file 1*.

## Generation of constructs and transgenic flies

Artificially synthesized $(G_4C_2)_{50}$ sequences flanked at the 5′ end with an *Eag*I recognition site and at the 3′ end with a *PspOM*I recognition site were subcloned into T-vector pMD20 (Takara Bio). To generate a longer repeat size, the pMD20-$(G_4C_2)_{50}$ vector was digested with *Eag*I and *PspOM*I, followed by ligation into the pMD20-$(G_4C_2)_{50}$ vector linearized by digestion with *Eag*I. This vector was digested with *EcoR*I and *Hind*III, and subcloned into the pcDNA3.1/*myc*-His(−)A vector (Thermo Fisher Scientific). We accidentally obtained the pcDNA3.1/*myc*-His(−)A-$(G_4C_2)_9$ vector at this step. The pcDNA3.1/*myc*-His(−)A-$(G_4C_2)_n$ vector was digested with *EcoR*I and *Xba*I, and subcloned into the *Drosophila* pUAST vector. These constructs have no start codon sequence (ATG) upstream of the $G_4C_2$ repeat sequence (*Figure 1—figure supplement 1A*). These pUAST-$(G_4C_2)_n$ vectors were amplified with a recombinase-mutated SURE2 *Escherichia coli* strain (Agilent Technologies) at 28°C for 72 hr to prevent repeat length contraction. The number of $G_4C_2$ repeats in the pUAST-$(G_4C_2)_{9 \text{ or } 50}$ vectors was determined by sequencing. To determine the number of $G_4C_2$ repeats in the pUAST-$(G_4C_2)_{89}$ vector, transposable element insertional mutagenesis using EZ-Tn5<KAN-2>Insertion Kit (Epicentre) and sequencing were performed. The entire sequence of the insert in the pUAST vector is shown in *Figure 1—figure supplement 1—source data 1*.

To generate pUAST-*FUS* or pUAST-*TDP-43* vectors, the Gateway Vector Conversion System (Thermo Fisher Scientific) was used. The human *FUS* or human *TARDBP* cDNA was subcloned into the pENTR/D-TOPO vector (Thermo Fisher Scientific). To generate the Gateway destination vector pUAST-DEST, we inserted the Gateway cassette A sequence (Thermo Fisher Scientific) into the pUAST vector. The pUAST-*FUS* or pUAST-*TDP-43* vectors were generated using Gateway recombination reactions (Thermo Fisher Scientific). The FUS RRM mutant construct (pUAST-*FUS-RRMmut*), in which leucine residues at positions 305, 341, 359, and 368 in the FUS protein were substituted to phenylalanine, was generated by PCR and the In-Fusion Cloning system (Takara Bio). To generate the other pUASTattB-*RBP* vectors, each cDNA encoding the RBP shown in *Figure 1—source data 1* and *Figure 6—source data 1* was subcloned into the pUASTattB vector (VectorBuilder). To establish transgenic flies harboring *UAS*-$(G_4C_2)_n$, *UAS-FUS*, *UAS-FUS* line 2, *UAS-FUS-RRMmut*, and *UAS-TDP-43*, the pUAST-$(G_4C_2)_n$, pUAST-*FUS*, pUAST-*FUS* line 2, pUAST-*FUS-RRMmut*, and pUAST-*TDP-43* vectors, respectively, were injected into fly embryos of the $w^{1118}$ strain. To establish transgenic flies harboring the other *UAS-RBP* constructs including *UAS-FUS* line 3, pUASTattB-*RBP* vectors were injected into fly embryos of the attP40 strain. By employing site-specific transgenesis using the pUASTattB vector, each transgene was inserted into the same locus of the genome and was expected to be expressed at the equivalent levels. These transgenic flies were established using standard methods at BestGene Inc.

The number of repeats in *UAS*-$(G_4C_2)_{9 \text{ or } 42}$ transgenic flies was determined by genomic PCR using the forward (5′-AACCAGCAACCAAGTAAATCAAC-3′) and reverse (5′-TGTTGAGAGTCAGCAGTAGCC-3′) primers, which amplifies a part of the *UAS*-$(G_4C_2)_n$ sequence, including $G_4C_2$ repeat sequence, followed by sequencing using the forward (5′-GCCAAGAAGTAATTATTGA-3′) and/or reverse (5′-TCCAATTATGTCACACC-3′) primers.

## Quantitative real-time PCR

Total RNA was extracted from female third-instar larvae of each genotype using TRIzol reagent (Thermo Fisher Scientific) according to the manufacturer's instructions. First-strand cDNA was synthesized using QuantiTect Reverse Transcription Kit (QIAGEN). Real-time PCR was performed using SYBR Premix Ex Taq II (Takara Bio) and the Mx3000P Real-time quantitative PCR system (Agilent Technologies) or the CFX96 Real-Time PCR Detection System (Bio-Rad). For $G_4C_2$ repeat RNA quantification of flies expressing $(G_4C_2)_n$ in *Figure 1—figure supplement 1C*, the forward (5′-ATGAATGGGAGCAGTGGTGG-3′) and reverse (5′-TGTTGAGAGTCAGCAGTAGCC-3′) primers were used. For $G_4C_2$ repeat RNA quantification of flies expressing $(G_4C_2)_{89}$(H) and FUS, FUS-RRMmut, other RNA-binding proteins, or *caz-IR* in *Figures 1E, 3C, 4G and 6D*, the forward (5′-CCCAATCCATATGACTAGTAGATCC-3′) and reverse (5′- TGTAGGTAGTTTGTCCAATTATGTCA-3′) primers were used. Both of the above-mentioned primer pairs recognize sequences downstream of the $G_4C_2$ repeats. For *gal4* mRNA quantification, the forward (5′-TTGAAATCGCGTCGAAGGA-3′) and reverse (5′-GGCTCCAATGGCTAATATGCA-3′) primers were used (*Li et al., 2008*). Data were analyzed using the standard curve method. The amounts of $G_4C_2$ repeat transcripts were normalized to those of *gal4* transcripts expressed in the

same tissue to avoid potential confounding derived from the difference in tissue viability between genotypes. At least three independent biological replicates per genotype were analyzed. Data were normalized by setting the values of the samples from flies expressing $(G_4C_2)_{89}$(H) (*Figure 1—figure supplement 1C*), both $(G_4C_2)_{89}$(H) and *EGFP* (*Figures 1E, 3C and 6D*), or both $(G_4C_2)_{89}$(H) and *GFP-IR* (*Figure 4G*) as 100.

## Imaging and quantification of fly eyes

Light microscopic images of the eyes of 3- to 5-day-old female flies were taken using a stereoscopic microscope model SZX10 (Olympus) with a CCD camera DP21 (Olympus). Images shown are representative eye phenotypes of the fly crosses. Crosses were performed three times to validate the specific phenotypes. Eye size and pigmentation were quantified as previously reported (*Saitoh et al., 2015*). 5 or 10 eyes per genotype were analyzed. Data were normalized by setting the values of samples from flies expressing one copy of *EGFP* (*Figure 4D*), those expressing two copies of *EGFP* (*Figures 1B–D, 2B, C, and 6B and C*, *Figure 1—figure supplement 2B–D*, and *Figure 4—figure supplement 1B–D*), or those expressing both *EGFP* and *GFP-IR* (*Figure 4B*), as 100.

## Egg-to-adult viability of flies

Mated female flies were placed on grape juice agar with yeast paste for 24 hr. Eggs were collected from the surface of the grape juice agar, and the number of eggs was counted and placed on new standard fly food. After eclosion, the number of adult flies was counted. Egg-to-adult viability was calculated by dividing the number of adult flies by the number of eggs. More than 500 eggs per genotype were used. Data were normalized by setting the values of samples from flies expressing two copies of EGFP as 100 (*Figure 2D*).

## Climbing assay

Twenty male flies were gently introduced into a glass vial. After a 5 min adaptation period, the bottom of the vial was gently tapped and the height the flies reached in 10 s was recorded using a digital video camera, and scored as follows: 0 (lower than 2 cm), 1 (from 2 to 3.9 cm), 2 (from 4 to 5.9 cm), 3 (from 6 to 7.9 cm), 4 (from 8 to 9.9 cm), and 5 (higher than 10 cm). Five trials were performed in each experiment at intervals of 20 s. The assay was performed between 8:00 and 10:00. Climbing scores were calculated as an average of five trials.

## GeneSwitch experiments

Flies were crossed in the absence of RU486 (mifepristone) on standard fly food. 1-day-old adult male flies were transferred to Formula 4-24 Instant *Drosophila* medium (Wako) with RU486 (100 µg/mL) for the indicated periods. Every 2 or 3 d, flies were transferred to new medium with RU486. Climbing assays were performed at 0, 7, and 14 d after the start of RU486 treatment (*Figure 2E*).

## RNA fluorescence in situ hybridization

Female third-instar larvae were dissected in ice-cold phosphate-buffered saline (PBS). Salivary glands were fixed with 4% paraformaldehyde (PFA) (pH 7.0) in PBS for 30 min and incubated in 100% methanol. Fixed samples were rehydrated in 75% (v/v), 50%, and 25% ethanol in PBS, and rinsed in PBS and distilled water (DW). Samples were then treated with 0.2 N HCl/DW for 20 min at room temperature (RT) and rinsed in DW. Next, the samples were permeabilized with 0.2% Triton X-100 in PBS for 10 min, rinsed in PBS for 5 min, fixed again in 4% PFA in PBS for 20 min, then washed twice for 5 min each in PBS, and incubated twice for 15 min each in 2 mg/mL glycine/PBS. After the acetylation treatment, samples were incubated for 1 hr at 37°C in hybridization buffer consisting of 50% formamide, 2× saline sodium citrate (SSC), 0.2 mg/mL yeast tRNA, and 0.5 mg/mL heparin. For hybridization, samples were incubated overnight at 80°C with a 5′ end Alexa 594-labeled $(G_2C_4)_4$ or Alexa 488-labeled $(C_2G_4)_4$ locked nucleic acid (LNA) probe (5 nM) in hybridization buffer. These LNA probes were synthesized by GeneDesign Inc. After the hybridization, samples were washed once for 5 min in 4× SSC at 80°C, three times for 20 min each in 2× SSC and 50% formamide at 80°C, three times for 40 min each in 0.1× SSC at 80°C, and once for 5 min in PBS containing 0.5% Triton X-100 (PBT) at RT. Nuclei were stained with 4',6-diamidino-2-phenylindole (DAPI) or 2'-(4-ethoxyphenyl)–5-(4-methyl-1-piperazinyl)–2,5'-bi-1-*H*-b enzimidazole, trihydrochloride (Hoechst 33342). Stained samples were mounted in SlowFade Gold

antifade reagent (Thermo Fisher Scientific) and observed under a Zeiss LSM710 or LSM880 confocal laser-scanning microscope.

After RNA FISH, samples were scanned using a Zeiss LSM710 or LSM880 confocal laser-scanning microscope along the z-axis direction. One z-stack image was taken per salivary gland using ZEN imaging software (Zeiss). RNA foci-positive nuclei in more than 30 cells per salivary gland were counted, and the percentage of nuclei containing RNA foci in the salivary gland was calculated. Ten salivary glands were analyzed for each genotype.

## Immunohistochemistry

Female third-instar larvae were dissected in ice-cold PBS. Eye imaginal discs and salivary glands were fixed with 4% PFA in PBS for 30 min and washed three times with PBT. After blocking with 5% goat serum/PBT, the samples were incubated overnight at 4°C with a rat monoclonal anti-poly(GR) antibody (clone 5A2, MABN778, Millipore), a mouse monoclonal anti-poly(GA) antibody (clone 5E9, MABN889, Millipore), a rabbit polyclonal anti-poly(GA) antibody (CAC-TIP-C9-P01, Cosmo Bio), a rabbit polyclonal anti-poly(GP) antibody (NBP2-25018, Novus Biologicals), or a rabbit polyclonal anti-FUS antibody (A300-302A, Bethyl Laboratories) at 1:1000 dilution as the primary antibody. After washing three times with PBT, the samples were incubated with an Alexa 633-conjugated anti-rat antibody (A-21094, Thermo Fisher Scientific), or an Alexa 488-conjugated or Alexa 555-conjugated anti-rabbit antibody (A-11008 or A-21428, respectively, Thermo Fisher Scientific) at 1:500 dilution as the secondary antibody. After washing three times with PBT, nuclei were stained with DAPI or Hoechst 33342. Stained samples were mounted in SlowFade Gold antifade reagent and observed using confocal laser-scanning microscopes (LSM710, LSM880 [Zeiss], and FV3000 [Olympus]).

The number of DPR aggregates in the eye discs was quantitatively measured using ZEN imaging software (Zeiss) and ImageJ (*Schneider et al., 2012*), as follows: (1) selection of photoreceptor neurons within the 13 developing ommatidia in rows 2 and 3 at the posterior end of the eye discs (*Saitoh et al., 2015*) by DAPI or Hoechst 33342 staining because these ommatidia are at similar stages of development and can be easily identified; and (2) counting of the number of DPR aggregates with a diameter of larger than 2 μm in the cytoplasm. 10–15 eye discs were analyzed for each genotype.

## Measurement of poly(GP) protein levels

The heads of 5-day-old female flies expressing both $(G_4C_2)_{89}$(H) and either *EGFP*, *FUS*, or *FUS-RRMmut* using the *GMR-Gal4* driver were collected and stored at –80°C. Samples were prepared using a previously reported method (*Tran et al., 2015*). Poly(GP) levels were measured by a sandwich immunoassay that uses Meso Scale Discovery electrochemiluminescence detection technology, as described previously (*Su et al., 2014*). Data were normalized by setting the values of samples from flies expressing $(G_4C_2)_{89}$(H) and *EGFP* (*Figure 3F*) as 100.

## Western blotting

To assess the expression levels of FUS and FUS-RRMmut (*Figure 2—figure supplement 1*), or GR-GFP (*Figure 3G–I*), 10 heads of 5-day-old female flies expressing *FUS* or *FUS-RRMmut*, or both *LDS-(G₄C₂)₄₄*^GR-GFP and either *FUS* or *FUS-RRMmut* using the *GMR-Gal4* driver were homogenized in 100 μL of sodium dodecyl sulfate (SDS) sample buffer using a pestle, boiled for 5 min, and centrifuged at $10,000 \times g$ for 3 min at 25°C. 5 μL of each supernatant were run on a 5–20% gradient polyacrylamide gel (Wako) and then transferred onto an Immun-Blot polyvinylidene fluoride membrane (Bio-Rad). Membranes were blocked with 5% skim milk in PBS containing 0.1% Tween-20 (PBST) or PVDF Blocking Reagent for Can Get Signal (TOYOBO) for 2 hr at RT, and then incubated overnight at 4°C with a rabbit polyclonal anti-FUS antibody (A300-302A, Bethyl Laboratories), a rat monoclonal anti-poly(GR) antibody (clone 5A2, MABN778, Millipore), a Living Colors EGFP mouse monoclonal antibody (632569, Clontech), or a mouse monoclonal anti-actin antibody (clone AC-40, A4700, Sigma-Aldrich) at 1:1000 dilution as the primary antibody. After washing three times with PBST, membranes were incubated for 2 hr at RT with either HRP-conjugated anti-rat, anti-rabbit, or anti-mouse antibody (31470, 31460, or 31430, respectively, Invitrogen) at 1:5000 dilution as the secondary antibody, washed three times with PBST, treated with SuperSignal West Dura chemiluminescent substrate (Thermo Fisher Scientific), and imaged using the LuminoGraphII imaging system (ATTO). Data were normalized by setting the

average values of samples from flies expressing *FUS* (*Figure 2—figure supplement 1B*) or those expressing both *LDS*-$(G_4C_2)_{44}$^GR-GFP and *DsRed* as 100 (*Figure 3H and I*).

## Filter binding assay

For preparation of FUS proteins with an N-terminal His tag (His-FUS and His-FUS-RRMmut), cDNAs of the human FUS protein (wild type or RRM mutant) from pUAST-*FUS* or pUAST-*FUS-RRMmut* were cloned into the multiple cloning site (*Xho*I and *Bam*HI) of the plasmid vector pET-15b (Novagen) (*Nomura et al., 2014*). After transfection of the plasmids into *E. coli* BL21 (DE3), the expression of His-FUS proteins was induced by culturing the transformed cells in the presence of 0.5 mM isopropyl β-*D*-thiogalactopyranoside (IPTG) at 20°C for 20 hr. Cells were lysed by ultrasonication in PBS (pH 7.4) containing 2% (v/v) Triton X-100, 1 M NaCl, DNase I, $MgSO_4$, and ethylenediaminetetraacetic acid-free cOmplete Protease Inhibitor Cocktail (Roche Applied Sciences). After centrifugation at $20,000 \times g$ for 30 min at 4°C, the pellets were redissolved in a buffer (pH 7.0) containing 6 M guanidine hydro-chloride (GdnHCl), 50 mM Tris, and 1 M NaCl. His-FUS proteins in the pellets were purified by $Ni^{2+}$ affinity chromatography. In brief, the His-FUS proteins were mixed with Profinity IMAC $Ni^{2+}$-charged resin (Bio-Rad) for 30 min at 20°C. Then, FUS proteins bound to the resin were washed with wash buffer (6 M GdnHCl, 50 mM Tris, and 1 M NaCl, pH 7.0), and eluted with elution buffer (6 M GdnHCl, 50 mM Tris, 100 mM NaCl, and 250 mM imidazole, pH 7.0). For preparation of soluble, refolded FUS, FUS proteins (200 μM) in the elution buffer were diluted 20-fold with a buffer (pH 7.0) containing 50 mM Tris, 100 mM NaCl, 10% (v/v) glycerol, and 5 mM Tris (2-carboxyethyl) phosphine (buffer A), which, however, produced significant amounts of precipitate. This insoluble material was removed by centrifugation at $20,000 \times g$ for 10 min at 4°C, resulting in the recovery of soluble FUS proteins in the supernatant fraction. Protein concentrations were spectroscopically determined from the absorbance at 280 nm using the following extinction coefficients: $71,630 \ cm^{-1} \ M^{-1}$ for both FUS and FUS-RRMmut.

Biotinylated RNAs were synthesized by GeneDesign Inc. 10 nM biotin-$(G_4C_2)_4$, 10 nM biotin-$(AAAAAA)_4$, or 10 nM biotin-$(UUAGGG)_4$ (telomeric repeat-containing RNA: TERRA) were incubated with soluble FUS proteins (5, 10, or 50 nM) in buffer A with 0.4 U/μL RNase inhibitor (RNasin Plus RNase Inhibitor, Promega). Biotinylated $(AAAAAA)_4$ and TERRA are negative and positive controls, respectively. After an hour at RT, the mixture was filtered through a nitrocellulose membrane (PROTRAN, 0.2 μm, Amersham Biosciences) overlaid on a nylon membrane (Hybond-N$^+$, 0.45 μm, Schleicher & Schuell) in a 96-well slot-blot apparatus (ATTO) (*Furukawa et al., 2011*). After extensive washing of the membranes with buffer A, the bound RNAs were crosslinked to the membranes using ultraviolet radiation (254 nm; UV Stratalinker, Stratagene) at an energy level of 0.12 J. After blocking with 3% (w/v) BSA in Tris-buffered saline with 0.1% Tween-20, the membranes were incubated with streptavi-din-HRP (1:5000; Nacalai Tesque), and the biotinylated RNAs on the membranes were detected with ImmunoStar LD reagent (Wako).

## Preparation of recombinant FUS protein

For preparation of the FUS proteins, the human *FUS* (WT) and *FUS-RRMmut* genes flanked at the 5′ end with an *Nde*I recognition site and at the 3′ end with a *Xho*I recognition site was amplified by PCR from pUAST-*FUS* and pUAST- *FUS-RRMmut*, respectively. PCR fragments were digested with *Nde*I and *Xho*I. These fragments were ligated into the cloning sites of the plasmid vector pET-21b (Novagen) between *Nde*I and *Xho*I. After transfection of the plasmids into *E. coli* BL21 (DE3), expression of the FUS protein was induced by culturing the transformed cells in the presence of 0.5 mM IPTG at 37°C for 6 hr. Cells were harvested by centrifugation and suspended with buffer B (10% glycerol, 20 mM 4-(2-hydroxyethyl)–1-piperazineethanesulfonic acid [HEPES]-NaOH [pH 7.0], 300 mM NaCl, 1 mM dithiothreitol [DTT], 1 mM ethylenediaminetetraacetic acid [EDTA], 0.1% Tween-20, and 0.1% benzamidine hydrochloride) containing 1.5 mg/mL lysozyme, and stored for 30 min on ice. Cell lysates were sonicated, and insoluble protein was collected by centrifugation. The pellet was solubilized in buffer C (6 M urea, 10% glycerol, 20 mM HEPES-NaOH [pH 7.0], 1 mM DTT, 1 mM EDTA, and 0.1% benzamidine hydrochloride). After centrifugation, supernatants were loaded onto a DE52 (GE Health-care) open column. The flow-through fraction was loaded onto a CM52 (GE Healthcare) open column. The flow-through fraction of DE52 was applied to a CAPTO S column (GE Healthcare), and the flow-through fraction was collected using the ÄKTAexplorer 10S/100 system (GE Healthcare). The flow-through fraction was applied to a Mono S column (GE Healthcare). Proteins were fractionated with a

0–500 mM linear gradient of NaCl in buffer D (6 M urea, 10% glycerol, 20 mM HEPES-NaOH [pH 7.0], 1 mM DTT, and 1 mM EDTA) using ÄKTA explorer 10S/100 system. The FUS fraction was eluted at 150–200 mM NaCl. For refolding, the eluted peak fraction was diluted fivefold using refolding buffer (900 mM arginine, 100 mM N-cyclohexyl-2-hydroxyl-3-aminopropanesulfonic acid [pH 9.5], 0.3 mM reduced glutathione, 0.03 mM oxidized glutathione, and 1 mM ZnCl$_2$), and stored overnight at RT. The solution was concentrated using a centrifugal filter (Vivaspin 6–10 kDa; GE Healthcare) to 1–2 mg/mL, and then dialyzed against buffer E (10% glycerol, 20 mM HEPES-NaOH [pH 6.8], 300 mM NaCl, 0.1 mM EDTA, and 10 mM β-cyclodextrin), and stored frozen at −80°C.

## Surface plasmon resonance analyses

The binding of FUS to $(G_4C_2)_4$ RNA was analyzed using a Biacore T200 instrument (GE Healthcare). $(G_4C_2)_4$ RNAs biotinylated at the 5′ end in 10 mM HEPES pH 6.8 and 500 mM MCl (M=K, Na, or Li) was injected over the streptavidin-coated surface of a sensor chip SA (GE Healthcare). The amount of immobilized RNA was as follows: 240 resonance unit (RU) in KCl, 363 RU in NaCl, or 319 RU in LiCl buffer condition. Binding experiments were performed using the single-cycle kinetics method. The running buffer used was 20 mM HEPES (pH 6.8), 1 mM MgCl$_2$, 0.05% Tween-20, and 150 mM KCl, NaCl, or LiCl. FUS was diluted in the running buffer and injected sequentially over the RNA-immobilized sensor surface in increasing concentrations (0.016, 0.031, 0.063, 0.13, or 0.25 µM). Sensorgrams were obtained at 25°C, 30 µL/min flow rate, 60 s of contact time, and 120 s of dissociation time.

## Circular dichroism spectroscopy

CD spectra were measured at 25°C using a spectropolarimeter model J-820 (JASCO). $(G_4C_2)_4$ RNA was synthesized by GeneDesign Inc and dissolved in 20 mM HEPES (pH 6), 18.75 mM NaCl, 10 mM MgCl$_2$, 0.625% glycerol, 0.625 mM β-cyclodextrin, and 0.0625 mM EDTA with 150 mM KCl, NaCl, or LiCl. RNA samples containing 150 mM KCl were first heated at 95°C for 5 min and then cooled to RT to form the G-quadruplex structure. The other samples were not heated. FUS (1 µM) was added to the RNA sample (4 µM) and mixed before recording the spectrum. CD spectra were recorded at a speed of 50 nm min$^{-1}$ and a resolution of 1 nm, and 10 scans were averaged.

## Nuclear magnetic resonance spectroscopy

All one-dimensional $^1$H NMR spectral data were recorded using AVANCE III 800 MHz NMR spectrometers equipped with a TXI cryogenic probe (Bruker BioSpin) at 25°C. $(G_4C_2)_4$ RNA dissolved in 20 mM HEPES (pH 6.8), 150 mM KCl, 18.75 mM NaCl, 10 mM MgCl$_2$, 0.625% glycerol, 0.625 mM β-cyclodextrin, and 0.0625 mM EDTA was first heated at 95°C for 5 min and then cooled to room temperature to form the G-quadruplex structure. The RNA (10 µM) was mixed with FUS at molar ratios (RNA:FUS) of 1:0, 1:0.2, 1:0.4, and 0:1. The samples were then prepared at a final concentration of 10% D$_2$O before recording their spectra. $^1$H NMR data were acquired using simple single 90° hard-pulse excitations following solvent signal suppression with a jump-and-return pulse scheme. Free induction decay data (1600 points in total) were collected by repeating the scans (2600 times) with an interscan delay of 2.5 s. All NMR data were processed using Topspin 3.6 software (Bruker BioSpin).

## RNA synthesis for in vitro translation

For preparation of the C9-RAN reporter plasmid, the pEF6-*C9orf72* intron1-$(G_4C_2)_{80}$ vector was digested with *HindIII* and *NotI* to obtain the fragment *C9orf72* intron1-$(G_4C_2)_{80}$ and subcloned into the pcDNA5/FRT vector (Thermo Fisher Scientific). To add the T7 promoter upstream of the *C9orf72* intron 1 sequence in this pcDNA5/FRT-*C9orf72* intron1-$(G_4C_2)_{80}$ vector, a forward primer including T7 promoter sequences with the 5′-terminal region of *C9orf72* intron 1 flanked at the 5′ end with an *HindIII* recognition site, and a reverse primer recognizing the 3′-terminal region of *C9orf72* intron 1 sequences including a *BssHII* recognition site were designed, and used to amplify a fragment containing *C9orf72* intron 1 with a T7 promoter by PCR. Then, this fragment was subcloned into the pcDNA5/FRT-*C9orf72* intron1-$(G_4C_2)_{80}$ vector digested by *HindIII* and *BssHII*. In addition to the T7 promoter, the Myc tag in the GA frame downstream of $(G_4C_2)_{80}$ was introduced into this vector.

The reporter plasmids were linearized with *XbaI*. Linearized DNA was in vitro transcribed using mMESSAGE mMACHINE T7 Transcription Kit (Invitrogen) according to the manufacturer's instructions. T7 reactions were carried out at 37°C for 2 hr, treated with TURBO DNaseI for 15 min at 37°C to

remove the DNA template, and then polyadenylated with *E. coli* Poly-A Polymerase (NEB) for 1 hr at 37°C. Synthesized mRNAs were purified by LiCl precipitation. The size and quality of the synthesized mRNAs were verified on a denaturing RNA gel.

### In vitro translation assay

mRNAs of *C9orf72* intron1-$(G_4C_2)_{80}$ with a Myc tag in the GA frame were in vitro translated with Flexi Rabbit Reticulocyte Lysate System (Promega) according to the manufacturer's instructions. Translation reactions were performed with 10 ng/µL mRNA and contained 30% rabbit reticulocyte lysate, 10 µM amino-acid mix minus methionine, 10 µM amino-acid mix minus leucine, 0.5 mM MgOAc, 100 mM KCl, and 0.8 U/µL Murine RNAse Inhibitor (NEB). FUS or FUS-RRMmut at each concentration (10, 100, 200, 400, and 1000 nM) was preincubated with mRNA for 10 min to facilitate the interaction between FUS protein and $G_4C_2$ repeat RNA, and added for translation in the lysate. Samples were incubated at 30°C for 90 min before termination by incubation on ice. 10 µL of samples were analyzed by 13% SDS-polyacrylamide gel electrophoresis and western blotting using a mouse monoclonal anti-c-Myc antibody (clone 9E10, Wako) as the primary antibody.

### Quantification and statistical analysis

Statistical parameters including the definitions and exact values of n (e.g. number of experiments, number of flies, number of eye imaginal discs, etc.), distributions, and deviations are stated in the figures and corresponding figure legends. Multiple-comparison tests using one-way ANOVA with Tukey's post hoc analysis were performed for *Figures 1B–E, 2B–D, 3B, C, E, F, H, I, 4B, D, and 6B–D, F*, *Figure 1—figure supplement 1C and D*, *Figure 1—figure supplement 2B–D*, and *Figure 4—figure supplement 1B–D*, multiple-comparison test using two-way repeated measures ANOVA with Tukey's post hoc analysis was performed for *Figure 2E*, and the unpaired *t*-test was used for *Figures 4F, G, I and 5G*, and *Figure 2—figure supplement 1B*. Differences in means were considered statistically significant at p<0.05. All statistical analyses were performed using GraphPad Prism version 8.3.4 (GraphPad Software, LLC).

As the sample sizes used in this study were similar to previous publications (*Freibaum et al., 2015*; *Goodman et al., 2019*; *Mizielinska et al., 2014*; *Xu et al., 2013*), statistical analyses were performed afterward without interim data analysis. Data were not excluded and were collected and processed randomly. Sample collection and analyses for the measurement of poly(GP) protein levels were performed in a double-blind manner. Data collection and analyses for other experiments were not performed in a blind manner regarding the conditions of the experiments.

## Acknowledgements

We thank Drs E Storkebaum (Radboud University), N Bonini (University of Pennsylvania), and B McCabe (École Polytechnique Fédérale de Lausanne) for kindly providing the *caz* mutant, *UAS-LDS-*$(G_4C_2)_{44}$^GR-GFP, and *UAS-FLAG-caz* and *UAS-FLAG-FUS* flies, respectively. We acknowledge Bloomington *Drosophila* Stock Center and Vienna *Drosophila* Resource Center for providing various fly stocks. We thank Drs H Imataka and K Machida (University of Hyogo) for kindly providing human tRNAs for in vitro translation. We also thank the members of our laboratory for helpful discussions, K Yamamoto, E Wakisaka, T Yamashita, and A Sugai for their technical assistance, Dr. H Akiko Popiel for critical reading and English editing of the manuscript, and the Center for Medical Research and Education, Graduate School of Medicine, Osaka University, for supplying experimental equipment.

## Additional information

### Competing interests

Morio Ueyama, Daisaku Ozawa, Tomoya Taminato, Toshihide Takeuchi, Yoshitaka Nagai: He previously belonged to the Department of Neurotherapeutics, Osaka University Graduate School of Medicine, that is an endowment department supported by Nihon Medi-Physics Co., AbbVie GK., Otsuka Pharm Co., Kyowakai Med. Co., Fujiikai Med. Co., Yukioka Hosp., Osaka Gyoumeikan Hosp., Kyorin Co., and Tokuyukai Med. Co. The other authors declare that no competing interests exist.

## Funding

| Funder | Grant reference number | Author |
|---|---|---|
| Ministry of Education, Culture, Sports, Science and Technology | Scientific Research on Innovative Areas (Brain Protein Aging and Dementia Control) 17H05699 | Yoshiaki Furukawa |
| Ministry of Education, Culture, Sports, Science and Technology | Scientific Research on Innovative Areas (Brain Protein Aging and Dementia Control) 17H05705 | Yoshiaki Furukawa |
| Ministry of Education, Culture, Sports, Science and Technology | Transformative Research Areas (A) (Multifaceted Proteins) 20H05927 | Yoshitaka Nagai Kohji Mori |
| Ministry of Education, Culture, Sports, Science and Technology | Strategic Research Program for Brain Sciences 11013026 | Yoshitaka Nagai |
| Japan Society for the Promotion of Science | Scientific Research (B) 21H02840 | Yoshitaka Nagai |
| Japan Society for the Promotion of Science | Scientific Research (B) 20H03602 | Kohji Mori |
| Japan Society for the Promotion of Science | Scientific Research (C) 15K09331 | Morio Ueyama |
| Japan Society for the Promotion of Science | Scientific Research (C) 19K07823 | Morio Ueyama |
| Japan Society for the Promotion of Science | Scientific Research (C) 17K07291 | Akira Ishiguro |
| Japan Society for the Promotion of Science | Young Scientists (A) 17H05091 | Kohji Mori |
| Japan Society for the Promotion of Science | Young Scientists (B) 25860733 | Morio Ueyama |
| Japan Society for the Promotion of Science | Challenging Exploratory Research 24659438 | Yoshitaka Nagai |
| Japan Society for the Promotion of Science | Challenging Exploratory Research 18K19515 | Kohji Mori |
| Ministry of Health, Labor and Welfare, Japan | Health Labor Sciences Research Grant for Research on Development of New Drugs H24-Soyaku-Sogo-002 | Yoshitaka Nagai |
| Japan Agency for Medical Research and Development | Strategic Research Program for Brain Sciences JP15dm0107026 | Yoshitaka Nagai |
| Japan Agency for Medical Research and Development | Strategic Research Program for Brain Sciences JP20dm0107061 | Yoshitaka Nagai |
| Japan Agency for Medical Research and Development | Practical Research Projects for Rare/Intractable Diseases JP16ek0109018 | Yoshitaka Nagai |
| Japan Agency for Medical Research and Development | Practical Research Projects for Rare/Intractable Diseases JP19ek0109222 | Yoshitaka Nagai |
| Japan Agency for Medical Research and Development | Practical Research Projects for Rare/Intractable Diseases JP20ek0109316 | Yoshitaka Nagai |

| Funder | Grant reference number | Author |
|---|---|---|
| Japan Agency for Medical Research and Development | Platform Project for Supporting Drug Discovery and Life Science Research JP19am0101072 | Yoshitaka Nagai |
| National Center of Neurology and Psychiatry | Intramural Research Grants for Neurological and Psychiatric Disorders 27-7 | Yoshitaka Nagai |
| National Center of Neurology and Psychiatry | Intramural Research Grants for Neurological and Psychiatric Disorders 27-9 | Yoshitaka Nagai |
| National Center of Neurology and Psychiatry | Intramural Research Grants for Neurological and Psychiatric Disorders 30-3 | Yoshitaka Nagai |
| National Center of Neurology and Psychiatry | Intramural Research Grants for Neurological and Psychiatric Disorders 30-9 | Yoshitaka Nagai |
| National Center of Neurology and Psychiatry | Intramural Research Grants for Neurological and Psychiatric Disorders 3-9 | Yoshitaka Nagai |
| Japan Amyotrophic Lateral Sclerosis Association | IBC Grant H28 | Yoshitaka Nagai |
| Takeda Science Foundation | 2017 | Morio Ueyama |
| Takeda Science Foundation | 2016 | Kohji Mori |
| SENSHIN Medical Research Foundation | 2018 | Kohji Mori |

The funders had no role in study design, data collection and interpretation, or the decision to submit the work for publication.

## Author contributions

Yuzo Fujino, Conceptualization, Data curation, Formal analysis, Validation, Investigation, Visualization, Writing – original draft, Writing – review and editing; Morio Ueyama, Conceptualization, Resources, Data curation, Formal analysis, Funding acquisition, Validation, Investigation, Visualization, Methodology, Writing – original draft; Taro Ishiguro, Conceptualization; Daisaku Ozawa, Toshihiko Sugiki, Asako Murata, Data curation, Formal analysis, Investigation, Visualization, Methodology; Hayato Ito, Eiichi Tokuda, Resources, Data curation, Formal analysis, Investigation, Visualization, Methodology; Akira Ishiguro, Resources, Data curation, Formal analysis, Funding acquisition, Investigation, Visualization, Methodology; Tania Gendron, Data curation, Formal analysis, Investigation, Methodology; Kohji Mori, Resources, Data curation, Formal analysis, Funding acquisition, Investigation, Methodology; Tomoya Taminato, Toshihide Takeuchi, Manabu Ikeda, Toshiki Mizuno, Hideki Mochizuki, Hidehiro Mizusawa, Keiji Wada, Kinya Ishikawa, Kazuhiko Nakatani, Leonard Petrucelli, Hideki Taguchi, Supervision; Takuya Konno, Akihide Koyama, Resources; Yuya Kawabe, Investigation; Yoshiaki Furukawa, Toshimichi Fujiwara, Supervision, Funding acquisition; Osamu Onodera, Resources, Supervision; Yoshitaka Nagai, Conceptualization, Supervision, Funding acquisition, Validation, Visualization, Methodology, Writing – original draft, Project administration, Writing – review and editing

## Author ORCIDs

Yuzo Fujino ![ORCID] http://orcid.org/0000-0002-1773-8998
Hayato Ito ![ORCID] http://orcid.org/0000-0002-1437-5853
Toshihiko Sugiki ![ORCID] http://orcid.org/0000-0003-1716-1241
Kohji Mori ![ORCID] http://orcid.org/0000-0003-2629-0723
Yuya Kawabe ![ORCID] http://orcid.org/0000-0002-5438-4797
Toshimichi Fujiwara ![ORCID] http://orcid.org/0000-0001-7739-3525
Osamu Onodera ![ORCID] http://orcid.org/0000-0003-3354-5472
Hideki Taguchi ![ORCID] http://orcid.org/0000-0002-6612-9339

Yoshitaka Nagai [ORCID] http://orcid.org/0000-0003-1037-3129

Reviewer #1 (Public Review): https://doi.org/10.7554/eLife.84338.3.sa1
Reviewer #2 (Public Review): https://doi.org/10.7554/eLife.84338.3.sa2
Reviewer #3 (Public Review): https://doi.org/10.7554/eLife.84338.3.sa3
Author Response: https://doi.org/10.7554/eLife.84338.3.sa4

## Additional files

**Supplementary files**
• Supplementary file 1. Full genotypes of the fly lines and their cultured temperatures.
• MDAR checklist

**Data availability**
All data generated or analysed during this study are included in the manuscript and supporting files.

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
