## [Editor Report · eLife assessment]

This **important** study demonstrates that the human FUS protein, which is implicated in ALS and related conditions, interacts with RNAs containing GGGGCC repeats and can regulate their translation by altering three-dimensional structures caused by these repeats. The study is carefully executed and the data provide **convincing** evidence for its major claims. This work will likely be of interest to researchers studying RNA binding proteins, and to those working on ALS and related diseases.

---

## [Referee Report · Reviewer #1 (Public Review)]

This is a carefully performed and well documented study to indicate that the FUS protein interacts with the GGGGCC repeat sequence in *Drosophila* fly models, and the mechanism appears to include modulating the repeat structure and mitigating RAN translation. They suggest FUS, as well as a number of other G-quadruplex binding RNA proteins, are RNA chaperones, meaning they can alter the structure of the expanded repeat sequence to modulate its biological activities.Overall this is a nicely done study with nice quantitation.

---

## [Referee Report · Reviewer #2 (Public Review)]

Fuijino et al provide interesting data describing the RNA-binding protein, FUS, for its ability to bind the RNA produced from the hexanucleotide repeat expansion of GGGGCC (G4C2). This binding correlates with reductions in RNA foci formation, the production of toxic dipeptides and concomitant reductions in toxic phenotypes seen in (G4C2)30+ expressing *Drosophila*. Both FUS and G4C2 repeats of >25 are associated with ALS/FTD spectrum disorders. Thus, these data are important for increasing our understanding of potential interactions between multiple disease genes.

---

## [Referee Report · Reviewer #3 (Public Review)]

In this manuscript Fujino and colleagues used C9-ALS/FTD fly models to demonstrate that FUS modulates the structure of (G4C2) repeat RNA as an RNA chaperone, and regulates RAN translation, resulting in the suppression of neurodegeneration in C9-ALS/FTD. They also confirmed that FUS preferentially binds to and modulates the G-quadruplex structure of (G4C2) repeat RNA, followed by the suppression of RAN translation. The potential significance of these findings is high, since C9ORF72 repeat expansion is the most common genetic cause of ALS/FTD, especially in Caucasian populations and the DPR proteins have been considered the major cause of the neurodegenerations.

1. While the effect of RBP as an RNA chaperone on (G4C2) repeat expansion is supposed to be dose-dependent according to (G4C2)n RNA expression, the first experiment of the screening for RBPs in C9-ALS/FTD flies lacks this concept. It is uncertain if the RBPs of the groups "suppression (weak)" and "no effect" were less or no ability of RNA chaperone or if the expression of the RBP was not sufficient, and if the RBPs of the group "enhancement" exacerbated the toxicity derived from (G4C2)89 RNA or the expression of the RBP was excessive. The optimal dose of any RBPs that bind to (G4C2) repeats may be able to neutralize the toxicity without the reduction of (G4C2)n RNA.

2. In relation to issue 1, the rescue effect of FUS on the fly expressing (G4C2)89 (FUS-4) in Figure 4-figure supplement 1 seems weaker than the other flies expressing both FUS and (G4C2)89 in Figure 1 and Figure 1-figure supplement 2. The expression level of both FUS protein and (G4C2)89 RNA in each line is important from the viewpoint of therapeutic strategy for C9-ALS/FTD.

3. While hallmarks of C9ORF72 are the presence of DPRs and the repeat-containing RNA foci, the loss of function of C9ORF72 is also considered to somehow contribute to neurodegeneration. It is unclear if FUS reduces not only the DPRs but also the protein expression of C9ORF72 itself.

4. In Figure 5E-F, it cannot be distinguished whether FUS binds to GGGGCC repeats or 5' flanking region. Same experiment should be done by using FUS-RRMmut to elucidate whether FUS binding is the major mechanism for this translational control. Authors should show that FUS binding to long GGGGCC repeats is important for RAN translation.

5. It is not possible to conclude, as the authors have, that G-quadruplex-targeting RBPs are generally important for RAN translation (Figure 6), without showing whether RBPs which do not affect to (G4C2)89 RNA levels lead to decreased DPR protein level or RNA foci.

---

## [Author Response]

The following is the authors' response to the original reviews.

**Reply to Public Reviews:**

**Reply to Reviewer #1:**
This is a carefully performed and well-documented study to indicate that the FUS protein interacts with the GGGGCC repeat sequence in *Drosophila* fly models, and the mechanism appears to include modulating the repeat structure and mitigating RAN translation. They suggest FUS, as well as a number of other G-quadruplex binding RNA proteins, are RNA chaperones, meaning they can alter the structure of the expanded repeat sequence to modulate its biological activities.

**Response:** We would like to thank the reviewer for her/his time for evaluating our manuscript. We are very happy to see the reviewer for highly appreciating our manuscript.

1. Overall this is a nicely done study with nice quantitation. It remains somewhat unclear from the data and discussions in exactly what way the authors mean that FUS is an RNA chaperone: is FUS changing the structure of the repeat or does FUS binding prevent it from folding into alternative in vivo structure?

**Response:** We appreciate the reviewer’s constructive comments. Indeed, we showed that FUS changes the higher-order structures of GGGGCC [G4C2] repeat RNA in vitro, and that FUS suppresses G4C2 RNA foci formation in vivo. According to the established definition of RNA chaperone, RNA chaperones are proteins changing the structures of misfolded RNAs without ATP use, resulting in the maintenance of proper RNAs folding (Rajkowitsich et al., 2007). Thus, we consider that FUS is classified into RNA chaperone. To clarify these interpretations, we revised the manuscript as follows.

(1) On page 10, line 215-219, the sentence “These results were in good agreement with our previous study on SCA31 showing the suppressive effects of FUS and other RBPs on RNA foci formation of UGGAA repeat RNA as RNA chaperones …” was changed to “These results were in good agreement with … RNA foci formation of UGGAA repeat RNA through altering RNA structures and preventing aggregation of misfolded repeat RNA as RNA chaperones …”.

(2) On page 17, line 363-366, the sentence “FUS directly binds to G4C2 repeat RNA and modulates its G-quadruplex structure, as evident by CD and NMR analyses (Figure 5), suggesting its functional role as an RNA chaperone.” was changed to “FUS directly binds to G4C2 repeat RNA and modulates its G-quadruplex structure as evident by CD and NMR analyses (Figure 5, Figure 5—figure supplement 2), and suppresses RNA foci formation in vivo (Figures 3A and 3B), suggesting its functional role as an RNA chaperone.”

**Reply to Reviewer #2:**
Fuijino et al. provide interesting data describing the RNA-binding protein, FUS, for its ability to bind the RNA produced from the hexanucleotide repeat expansion of GGGGCC (G4C2). This binding correlates with reductions in the production of toxic dipeptides and reductions in toxic phenotypes seen in (G4C2)30+ expressing *Drosophila*. Both FUS and G4C2 repeats of >25 are associated with ALS/FTD spectrum disorders. Thus, these data are important for increasing our understanding of potential interactions between multiple disease genes. However, further validation of some aspects of the provided data is needed, especially the expression data.

**Response:** We would like to thank the reviewer for her/his time for evaluating our manuscript and also for her/his important comments that helped to strengthen our manuscript.

Some points to consider when reading the work:1. The broadly expressed GMR-GAL4 driver leads to variable tissue loss in different genotypes, potentially confounding downstream analyses dependent on viable tissue/mRNA levels.

**Response:** We thank the reviewer for this constructive comment. In the RT-qPCR experiments (Figures 1E, 3C, 4G, 6D and Figure 1—figure supplement 1C), the amounts of G4C2 repeat transcripts were normalized to those of *gal4* transcripts expressed in the same tissue, to avoid potential confounding derived from the difference in tissue viability between genotypes, as the reviewer pointed out. To clarify this process, we have made the following change to the revised manuscript.

(1) On page 30, line 548-550, the sentence “The amounts of G4C2 repeat transcripts were normalized to those of *gal4* transcripts in the same sample” was changed to “The amounts of G4C2 repeat transcripts were normalized to those of *gal4* transcripts expressed in the same tissue to avoid potential confounding derived from the difference in tissue viability between genotypes”.

2. The relationship between FUS and foci formation is unclear and should be interpreted carefully.

**Response:** We appreciate the reviewer’s important comment. We apologize for the lack of clarity. We showed the relationship between FUS and RNA foci formation in our C9-ALS/FTD fly, that is, FUS suppresses RNA foci formation (Figures 3A and 3B), and knockdown of endogenous *caz*, a *Drosophila* homologue of *FUS*, enhanced it conversely (Figures 4E and 4F). We consider that FUS suppresses RNA foci formation through altering RNA structures and preventing aggregation of misfolded G4C2 repeat RNA as an RNA chaperone. To clarify these interpretations, we revised the manuscript as follows.

(1) On page 10, line 215-219, the sentence “These results were in good agreement with our previous study on SCA31 showing the suppressive effects of FUS and other RBPs on RNA foci formation of UGGAA repeat RNA as RNA chaperones …” was changed to “These results were in good agreement with … RNA foci formation of UGGAA repeat RNA through altering RNA structures and preventing aggregation of misfolded repeat RNA as RNA chaperones …”.

(2) On page 17, line 363-366, the sentence “FUS directly binds to G4C2 repeat RNA and modulates its G-quadruplex structure, as evident by CD and NMR analyses (Figure 5), suggesting its functional role as an RNA chaperone.” was changed to “FUS directly binds to G4C2 repeat RNA and modulates its G-quadruplex structure as evident by CD and NMR analyses (Figure 5, Figure 5—figure supplement 2), and suppresses RNA foci formation in vivo (Figures 3A and 3B), suggesting its functional role as an RNA chaperone.”

**Reply to Reviewer #3:**
In this manuscript Fujino and colleagues used C9-ALS/FTD fly models to demonstrate that FUS modulates the structure of (G4C2) repeat RNA as an RNA chaperone, and regulates RAN translation, resulting in the suppression of neurodegeneration in C9-ALS/FTD. They also confirmed that FUS preferentially binds to and modulates the G-quadruplex structure of (G4C2) repeat RNA, followed by the suppression of RAN translation. The potential significance of these findings is high since C9ORF72 repeat expansion is the most common genetic cause of ALS/FTD, especially in Caucasian populations and the DPR proteins have been considered the major cause of the neurodegenerations.

**Response:** We would like to thank the reviewer for her/his time for evaluating our manuscript. We are grateful to the reviewer for the insightful comments, which were very helpful for us to improve the manuscript.

1. While the effect of RBP as an RNA chaperone on (G4C2) repeat expansion is supposed to be dose-dependent according to (G4C2)n RNA expression, the first experiment of the screening for RBPs in C9-ALS/FTD flies lacks this concept. It is uncertain if the RBPs of the groups "suppression (weak)" and "no effect" were less or no ability of RNA chaperone or if the expression of the RBP was not sufficient, and if the RBPs of the group "enhancement" exacerbated the toxicity derived from (G4C2)89 RNA or the expression of the RBP was excessive. The optimal dose of any RBPs that bind to (G4C2) repeats may be able to neutralize the toxicity without the reduction of (G4C2)n RNA.

**Response:** We appreciate the reviewer’s constructive comments. We employed the site-directed transgenesis for the establishment of RBP fly lines, to ensure the equivalent expression levels of the inserted transgenes. We also evaluated the toxic effects of overexpressed RBPs themselves by crossbreeding with control *EGFP* flies, showing in Figure 1A. To clarify them, we have made the following changes to the revised manuscript.

(1) On page 8, line 166-168, the sentence “The variation in the effects of these G4C2 repeat-binding RBPs on G4C2 repeat-induced toxicity may be due to their different binding affinities to G4C2 repeat RNA, and their different roles in RNA metabolism.” was changed to “The variation in the effects of these G4C2 repeat-binding RBPs on G4C2 repeat-induced toxicity may be due to their different binding affinities to G4C2 repeat RNA, and the different toxicity of overexpressed RBPs themselves.”.

(2) On page 29, line 519-522, the sentence “By employing site-specific transgenesis using the pUASTattB vector, each transgene was inserted into the same locus of the genome, and was expected to be expressed at the equivalent levels.” was added.

2. In relation to issue 1, the rescue effect of FUS on the fly expressing (G4C2)89 (FUS-4) in Figure 4-figure supplement 1 seems weaker than the other flies expressing both FUS and (G4C2)89 in Figure 1 and Figure 1-figure supplement 2. The expression level of both FUS protein and (G4C2)89 RNA in each line is important from the viewpoint of therapeutic strategy for C9-ALS/FTD.

**Response:** We appreciate the reviewer’s important comment. The *FUS-4* transgene is expected to be expressed at the equivalent level to the *FUS-3* transgene, since they are inserted into the same locus of the genome by the site-directed transgenesis. Thus, we suppose that the weaker suppressive effect of *FUS-4* coexpression on G4C2 repeat-induced eye degeneration can be attributed to the C-terminal FLAG tag that is fused to FUS protein expressed in *FUS-4* fly line. Since the *caz* fly expresses caz protein also fused to FLAG tag at the C-terminus, we used this *FUS-4* fly line to directly compare the effect of caz on G4C2 repeat-induced toxicity to that of FUS.

3. While hallmarks of C9ORF72 are the presence of DPRs and the repeat-containing RNA foci, the loss of function of C9ORF72 is also considered to somehow contribute to neurodegeneration. It is unclear if FUS reduces not only the DPRs but also the protein expression of C9ORF72 itself.

**Response:** We thank the reviewer for this comment. We agree that not only DPRs, but also toxic repeat RNA and the loss-of-function of C9ORF72 jointly contribute to the pathomechanisms of C9-ALS/FTD. Since *Drosophila* has no homolog corresponding to the human *C9orf72* gene, the effect of FUS on *C9orf72* expression cannot be assessed. Our fly models are useful for evaluating gain-of-toxic pathomechanisms such as RNA foci formation and RAN translation, and the association between FUS and loss-of function of C9ORF72 is beyond the scope of this study.

4. In Figure 5E-F, it cannot be distinguished whether FUS binds to GGGGCC repeats or the 5' flanking region. The same experiment should be done by using FUS-RRMmut to elucidate whether FUS binding is the major mechanism for this translational control. Authors should show that FUS binding to long GGGGCC repeats is important for RAN translation.

**Response:** We would like to thank the reviewer for these insightful comments. Following the reviewer’s suggestion, we perform in vitro translation assay again using FUS-RRMmut, which loses the binding ability to G4C2 repeat RNA as evident by the filter binding assay (Figure 5A), instead of BSA. The results are shown in the figures of Western blot analysis below. The addition of FUS to the translation system suppressed the expression levels of GA-Myc efficiently, whereas that of FUS-RRMmut did not. FUS decreased the expression level of GA-Myc at as low as 10nM, and nearly eliminated RAN translation activity at 100nM. At 400nM, FUS-RRMmut weakly suppressed the GA-Myc expression levels probably because of the residual RNA-binding activity. These results suggest that FUS suppresses RAN translation in vitro through direct interactions with G4C2 repeat RNA.

Unfortunately, RAN translation from short G4C2 repeat RNA was not investigated in our translation system, although the previous study reported the low efficacy of RAN translation from short G4C2 repeat RNA (Green et al., 2017).

**Author response image 1.**

(A) Western blot analysis of the GA-Myc protein in the samples from in vitro translation.

(B) Quantification of the GA-Myc protein levels.

We have made the following changes to the revised manuscript.

(1) Figure 5F was replaced to new Figures 5F and 5G.

(2) On page 14-15, line 326-330, the sentence “Notably, the addition of FUS to this system decreased the expression level of GA-Myc in a dose-dependent manner, whereas the addition of the control bovine serum albumin (BSA) did not (Figure 5F).” was changed to “Notably, upon the addition to this translation system, FUS suppressed RAN translation efficiently, whereas FUS-RRMmut did not. FUS decreased the expression levels of GA-Myc at as low as 10nM, and nearly eliminated RAN translation activity at 100nM. At 400nM, FUS-RRMmut weakly suppressed the GA-Myc expression levels probably because of the residual RNA-binding activity (Figure 5F and 5G).”.

(3) On page 15, line 330-332, the sentence “Taken together, these results indicate that FUS suppresses RAN translation from G4C2 repeat RNA in vitro as an RNA chaperone.” was changed to “Taken together, these results indicate that FUS suppresses RAN translation in vitro through direct interactions with G4C2 repeat RNA as an RNA chaperone.”.

(4) On page 37, line 720-723, the sentence “For preparation of the FUS protein, the human *FUS* (WT) gene flanked at the 5¢ end with an _Nde_I recognition site and at the 3¢ end with a _Xho_I recognition site was amplified by PCR from pUAST-*FUS*.” was changed to “For preparation of the FUS proteins, the human *FUS* (WT) and *FUS-RRMmut* genes flanked at the 5¢ end with an _Nde_I recognition site and at the 3¢ end with a _Xho_I recognition site was amplified by PCR from pUAST-*FUS* and pUAST- *FUS-RRMmut*, respectively.”.

(5) On page 41, line 816-819, the sentence “FUS or BSA at each concentration (10, 100, and 1,000 nM) was added for translation in the lysate.” was changed to “FUS or FUS-RRMmut at each concentration (10, 100, 200, 400, and 1,000 nM) was preincubated with mRNA for 10 min to facilitate the interaction between FUS protein and G4C2 repeat RNA, and added for translation in the lysate.”.

5. It is not possible to conclude, as the authors have, that G-quadruplex-targeting RBPs are generally important for RAN translation (Figure 6), without showing whether RBPs that do not affect (G4C2)89 RNA levels lead to decreased DPR protein level or RNA foci.

**Response:** We appreciate the reviewer’s critical comment. Following the suggestion by the reviewer, we evaluate the effect of these G-quadruplex-targeting RBPs on RAN translation. We additionally performed immunohistochemistry of the eye imaginal discs of fly larvae expressing (*G4C2*)89 and these G-quadruplex-targeting RBPs. As shown in the figures of immunohistochemistry below, we found that coexpression of *EWSR1*, *DDX3X*, *DDX5*, and *DDX17* significantly decreased the number of poly(GA) aggregates. The results suggest that these G-quadruplex-targeting RBPs regulate RAN translation as well as FUS.

**Author response image 2.**

(A) Immunohistochemistry of poly(GA) in the eye imaginal discs of fly larvae expressing (*G4C2*)89 and the indicated G-quadruplex-targeting RBPs.

(B) Quantification of the number of poly(GA) aggregates.

We have made the following changes to the revised manuscript.

(1) Figures 6E and 6F were added.

(2) On page 6-7, line 135-137, the sentence “In addition, other G-quadruplex-targeting RBPs also suppressed G4C2 repeat-induced toxicity in our C9-ALS/FTD flies.” was changed to “In addition, other G-quadruplex-targeting RBPs also suppressed RAN translation and G4C2 repeat-induced toxicity in our C9-ALS/FTD flies.”.

(3) On page 15, line 344-346, the sentence “As expected, these RBPs also decreased the number of poly(GA) aggregates in the eye imaginal discs (Figures 6E and 6F).” was added.

(4) On page 15, line 346-347, the sentence “Their effects on G4C2 repeat-induced toxicity and repeat RNA expression were consistent with those of FUS.” was changed to “Their effects on G4C2 repeat-induced toxicity, repeat RNA expression, and RAN translation were consistent with those of FUS.”

(5) On page 16, line 355-357, the sentence “Thus, some G-quadruplex-targeting RBPs regulate G4C2 repeat-induced toxicity by binding to and possibly by modulating the G-quadruplex structure of G4C2 repeat RNA.” was changed to “Thus, some G-quadruplex-targeting RBPs regulate RAN translation and G4C2 repeat-induced toxicity by binding to and possibly by modulating the G-quadruplex structure of G4C2 repeat RNA.”

(6) On page 19, line 417-421, the sentence “We further found that G-quadruplex-targeting RNA helicases, including DDX3X, DDX5, and DDX17, which are known to bind to G4C2 repeat RNA (Cooper-Knock et al., 2014; Haeusler et al., 2014; Mori et al., 2013a; Xu et al., 2013), also alleviate G4C2 repeat-induced toxicity without altering the expression levels of G4C2 repeat RNA in our *Drosophila* models.” was changed to “We further found that G-quadruplex-targeting RNA helicases, … ,also suppress RAN translation and G4C2 repeat-induced toxicity without altering the expression levels of G4C2 repeat RNA in our *Drosophila* models.”.

**Reply to Recommendations For The Authors:**
1. It is not clear from the start that the flies they generated with the repeat have an artificial vs human intronic sequence ahead of the repeat. It would be nice if they presented somewhere the entire sequence of the insert. The reason being that it seems they also tested flies with the human intronic sequence, and the effect may not be as strong (line 234). In any case, in the future, with a new understanding of RAN translation, it would be nice to compare different transgenes, and so as much transparency as possible would be helpful regarding sequences. Can they include these data?

**Response:** We thank the editors and reviewers for this comment. We apologize for the lack of clarity. We used artificially synthesized G4C2 repeat sequences when generating constructs for (*G4C2*)n transgenic flies, so these constructs do not contain human intronic sequence ahead of the G4C2 repeat in the *C9orf72* gene, as explained in the Materials and Methods section. To clarify the difference between our C9-ALS/FTD fly models and LDS-(*G4C2*)44GR-GFP fly model (Goodman et al., 2019), we have made the following change to the revised manuscript.

(1) Schema of the LDS-(*G4C2*)44GR-GFP construct was presented in Figure 3—figure supplement 1.

Furthermore, to maintain transparency of the study, we have provided the entire sequence of the insert as the following source file.

(2) The artificial sequences inserted in the pUAST vector for generation of the (*G4C2*)n flies were presented in Figure 1—figure supplement 1—source data 1.

2. It is really nice how they quantitated everything and showed individual data points.

**Response:** We thank the editors and reviewers for appreciating our data analysis method. All individual data points and statistical analyses are summarized in source data files.

3. So when they call FUS an RNA chaperone, are they simply meaning it is changing the structure of the repeat, or could it just be interacting with the repeat to coat the repeat and prevent it from folding into whatever in vivo structures? Can they speculate on why some RNA chaperones lead to presumed decay of the repeat and others do not? Can they discuss these points in the discussion? Detailed mechanistic understanding of RNA chaperones that ultimately promote decay of the repeat might be of highly significant therapeutic benefit.

**Response:** We appreciate these critical comments. Indeed, we showed that FUS changes the higher-order structures of G4C2 repeat RNA in vitro, and that FUS suppresses G4C2 RNA foci formation. According to the established definition of RNA chaperone, RNA chaperones are proteins changing the structures of misfolded RNAs without ATP use, resulting in the maintenance of proper RNAs folding (Rajkowitsich et al., 2007). Thus, we consider that FUS is classified into RNA chaperone. To clarify these interpretations, we revised the manuscript as follows.

(1) On page 10, line 215-219, the sentence “These results were in good agreement with our previous study on SCA31 showing the suppressive effects of FUS and other RBPs on RNA foci formation of UGGAA repeat RNA as RNA chaperones …” was changed to “These results were in good agreement with … RNA foci formation of UGGAA repeat RNA through altering RNA structures and preventing aggregation of misfolded repeat RNA as RNA chaperones …”.

(2) On page 17, line 363-366, the sentence “FUS directly binds to G4C2 repeat RNA and modulates its G-quadruplex structure, as evident by CD and NMR analyses (Figure 5), suggesting its functional role as an RNA chaperone.” was changed to “FUS directly binds to G4C2 repeat RNA and modulates its G-quadruplex structure as evident by CD and NMR analyses (Figure 5, Figure 5—figure supplement 2), and suppresses RNA foci formation in vivo (Figures 3A and 3B), suggesting its functional role as an RNA chaperone.”

Besides these RNA chaperones, we observed the expression of IGF2BP1, hnRNPA2B1, DHX9, and DHX36 decreased G4C2 repeat RNA expression levels. In addition, we recently reported that hnRNPA3 reduces G4C2 repeat RNA expression levels, leading to the suppression of neurodegeneration in C9-ALS/FTD fly models (Taminato et al., 2023). We speculate these RBPs could be involved in RNA decay pathways as components of the P-body or interactors with the RNA deadenylation machinery (Tran et al., 2004; Katahira et al., 2008; Geissler et al., 2016; Hubstenberger et al., 2017), possibly contributing to the reduced expression levels of G4C2 repeat RNA. To clarify these interpretations, we revised the manuscript as follows.

(3) On page 18, line 392-398, the sentences “Similarly, we recently reported that hnRNPA3 reduces G4C2 repeat RNA expression levels, leading to the suppression of neurodegeneration in C9-ALS/FTD fly models (Taminato et al., 2023). Interestingly, these RBPs have been reported to be involved in RNA decay pathways as components of the P-body or interactors with the RNA deadenylation machinery (Tran et al., 2004; Katahira et al., 2008; Geissler et al., 2016; Hubstenberger et al., 2017), possibly contributing to the reduced expression levels of G4C2 repeat RNA.” was added.

4. What is the level of the G4C2 repeat when they knock down caz? Is it possible that knockdown impacts the expression level of the repeat? Can they show this (or did they and I miss it)?

**Response:** We thank the editors and reviewers for this comment. The expression levels of G4C2 repeat RNA in (*G4C2*)89 flies were not altered by the knockdown of *caz*, as shown in Figure 4G.

5. A puzzling point is that FUS is supposed to be nuclear, so where is FUS in the brain in their lines? They suggest it modulates RAN translation, and presumably, that is in the cytoplasm. Is FUS when overexpressed now in part in the cytoplasm? Is the repeat dragging it into the cytoplasm? Can they address this in the discussion? If FUS is never found in vivo in the cytoplasm, then it raises the point that the impact they find of FUS on RAN translation might not reflect an in vivo situation with normal levels of FUS.

**Response:** We appreciate these important comments. We agree with the editors and reviewers that FUS is mainly localized in the nucleus. However, FUS is known as a nucleocytoplasmic shuttling RBP that can transport RNA into the cytoplasm. Indeed, FUS is reported to facilitate transport of actin-stabilizing protein mRNAs to function in the cytoplasm (Fujii et al., 2005). Thus, we consider that FUS binds to G4C2 repeat RNA in the cytoplasm and suppresses RAN translation in this study.

6. When they are using 2 copies of the driver and repeat, are they also using 2 copies of FUS? These are quite high levels of transgenes.

**Response:** We thank the editors and reviewers for this comment. We used only 1 copy of *FUS* when using 2 copies of *GMR-Gal4* driver. Full genotypes of the fly lines used in all experiments are described in Supplementary file 1.

7. In Figure5-S1, FUS colocalizing with (G4C2)RNA is not clear. High-magnification images are recommended.

**Response:** We appreciate this constructive comment on the figure. Following the suggestion, high-magnification images are added in Figure 5—figure supplement 1.

8. I also suggest that the last sentence of the Discussion be revised as follows: Thus, our findings contribute not only to the elucidation of C9-ALS/FTD, but also to the elucidation of the repeat-associated pathogenic mechanisms underlying a broader range of neurodegenerative and neuropsychiatric disorders than previously thought, and it will advance the development of potential therapies for these diseases.

**Response:** We appreciate this recommendation. We have made the following change based on the suggested sentence.

(1) On page 20-21, line 455-459, “Thus, our findings contribute not only towards the elucidation of repeat-associated pathogenic mechanisms underlying a wider range of neuropsychiatric diseases than previously thought, but also towards the development of potential therapies for these diseases.” was changed to “Thus, our findings contribute to the elucidation of the repeat-associated pathogenic mechanisms underlying not only C9-ALS/FTD, but also a broader range of neuromuscular and neuropsychiatric diseases than previously thought, and will advance the development of potential therapies for these diseases.”.

**Authors’ comment on previous eLife assessment:**

We thank the editors and reviewers for appreciating our study. We mainly evaluated the function of human FUS protein on RAN translation and G4C2 repeat-induced toxicity using *Drosophila* expressing human *FUS in vivo*, and the recombinant human FUS protein *in vitro*. To validate that FUS functions as an endogenous regulator of RAN translation, we additionally evaluated the function of *Drosophila* caz protein as well. We are afraid that the first sentence of the eLife assessment, that is, “This important study demonstrates that the Drosophila FUS protein, the human homolog of which is implicated in amyotrophic lateral sclerosis (ALS) and related conditions, …” is somewhat misleading. We would be happy if you modify this sentence like “This important study demonstrates that the human FUS protein, which is implicated in amyotrophic lateral sclerosis (ALS) and related conditions, …”.